



# Absorption closure in highly aged biomass burning smoke

Jonathan W. Taylor[1], Huihui Wu[1], Kate Szpek[3], Keith Bower[1], Ian Crawford[1], Michael J. Flynn[1], Paul I. Williams[1,2], James Dorsey[1,2], Justin M. Langridge[3], Michael I. Cotterell[4,*], Cathryn Fox[3], Nicholas W. Davies[3,4], Jim M. Haywood[3,4], and Hugh Coe[1]

[1]Centre for Atmospheric Science, Department of Earth and Environmental Sciences, University of Manchester, United Kingdom
[2]National Centre for Atmospheric Science, University of Manchester, United Kingdom
[3]Met Office, Exeter, United Kingdom
[4]College for Engineering, Mathematics and Physical Sciences, University of Exeter, Exeter, United Kingdom
[*]Now at School of Chemistry, University of Bristol, Bristol, United Kingdom

**Correspondence:** Jonathan Taylor (jonathan.taylor@manchester.ac.uk)

**Abstract.** The optical properties of black carbon (BC) are a major source of uncertainty in regional and global climate studies. In the past, detailed investigation of BC absorption has been hampered by systematic biases in the measurement instrumentation. We present airborne measurements of aerosol absorption and black carbon microphysical properties in highly aged biomass burning plumes measured over the southeast Atlantic ocean during CLARIFY-2017, using a suite of novel photoacoustic spectrometers to measure aerosol absorption at 405 nm, 514 nm, and 655 nm, and a single-particle soot photometer to measure the BC mass concentration, size, and mixing state. These measurements are of sufficient quality and detail to provide constraint on optical schemes used in climate models for the first time in biomass burning plumes far from source, an aerosol environment that is one of the most important climatically.

The average absorption Angstrom exponents (AAE) were 1.39 over the wavelength range 405 – 514 nm, and 0.94 over the range 514 – 655 nm, suggesting brown carbon (BrC) contributed to $10 \pm 2\%$ of absorption at 405 nm. The effective OA mass absorption coefficient (MAC) was $0.27 \pm 0.08 \, \mathrm{m^2 \, g^{-1}}$ at 405 nm. The BC particles were universally thickly-coated, and almost no externally-mixed BC particles were detected. The MAC of BC was also high, with equivalent absorption enhancements of around 1.8 at all three wavelengths, suggesting that the thick coatings acted as a lens that enhanced light absorption by the BC.

We compared the measured MAC and AAE values with those calculated using several optical models and absorption parametrisations that took the measured BC mass and mixing state as inputs. Homogeneous grey sphere Mie models were only able to replicate MAC for some low (real and imaginary) values of the complex BC refractive index ($m_{BC}$) at the shortest wavelength, but they would have to use unrealistically low values of $m_{BC}$ to accurately replicate AAE. A core/shell Mie model was able to generate good agreement for MAC in the green/red end of the visible spectrum for most values of $m_{BC}$. However, there are no possible values of $m_{BC}$ that produce MAC values that agree with our observations at all three wavelengths, due to a wavelength-dependent underestimation of the MAC of the underlying BC core. Four semi-empirical parametrisations from literature were also tested, linking BC mixing state to either MAC or absorption enhancement. Two of these schemes produced results that agreed within a few percent of the measured MAC at all three wavelengths, and AAE agreed well when discounting the effects of BrC.





Our results uniquely demonstrate the validity of absorption parametrisations, as well as the failings of Mie calculations, in this highly aged environment. We recommend future work should conduct similar analyses in environments where BC has different properties, and investigate the impact of implementing these types of schemes within climate models, as well as developing equivalent schemes for light scattering by soot particles at visible wavelengths.

## 1  Introduction

Every year, vast plumes of smoke are lofted into the free troposphere by open biomass burning in central and southern Africa. These plumes make their way westward over the ocean, and have an important effect on the radiative budget over the southeast Atlantic. Near the African continent, a stratocumulus deck sits atop the boundary layer, presenting a high-albedo surface that reflects solar radiation. Further west, the boundary layer deepens and the cloud deck becomes more broken, revealing the low-albedo sea surface below. Over cloud, the presence of absorbing aerosol in the free troposphere lowers the planetary albedo, but can potentially thicken the cloud deck by warming the free troposphere and strengthening the trade inversion (Wilcox, 2010). Smoke is also entrained into the boundary layer, where it increases concentrations of cloud condensation nuclei (CCN), brightening the clouds and raising their albedo, but this entrainment may also reduce cloud cover by warming the boundary layer and weakening the inversion (Zhang and Zuidema, 2019). These counterbalancing effects are all sensitive to the properties of the aerosol, and the radiative effects are particularly sensitive to the aerosol optical properties. More absorbing aerosol could change the sign of the associated radiative forcing for the semi-direct effect from negative to positive (Zhou et al., 2017).

Black carbon (BC) aerosol is the main absorbing component in these smoke plumes, and is a major climate warming agent globally. Open biomass burning is responsible for around 40% of all BC emissions, and African fires make up around 40% of all open burning (Bond et al., 2013). The mass absorption coefficient (MAC, the ratio of absorption cross-section to BC particle mass) is the key variable for characterising the absorbing properties of BC. The MAC of fresh, uncoated BC is relatively well constrained; Bond and Bergstrom (2006) summarised measurements and found the MAC of uncoated BC to be $7.5 \pm 1.2 \, \mathrm{m^2 \, g^{-1}}$ at a wavelength of $550 \, \mathrm{nm}$, with a wavelength-dependence that can be accurately described using an absorption Angstrom exponent (AAE) of 1. To date this work remains the most comprehensive summary of the MAC of fresh soot, and scientific effort since then has mostly focused on the modification of BC's absorbing properties by non-BC species such as organic aerosol (OA) and inorganic salts (e.g. Liu et al., 2017), as well as absorption by brown carbon (BrC) (e.g. Forrister et al., 2015; Healy et al., 2015). Where BC is encapsulated by non-BC material, the MAC of the soot may increase by a lensing effect, which is often quantified using absorption enhancement ($E_{\mathrm{Abs}}$, the ratio of the absorption of coated BC to that of uncoated BC). In biomass burning plumes, BC demonstrates some internal mixing in the first few hours after emission (Akagi et al., 2012), but the effects of this mixing in terms of optical properties remain poorly quantified over timescales of both hours and days.

The core/shell Mie model considers encapsulated BC as two concentric spheres, and predicts thickly coated particles can have absorption enhanced by a factor of $2 - 3$ (Bond et al., 2006). However, $E_{\mathrm{Abs}}$ values in this range have almost exclusively been measured where particles have been artificially aged (Schnaiter et al., 2005; Mikhailov et al., 2006; Peng et al., 2016), and





have not generally been found in atmospheric measurements. Several studies have shown that using the Mie core/shell model can overestimate $E_{\text{Abs}}$ (e.g. Cappa et al., 2012; Healy et al., 2015). Liu et al. (2017) summarised measurements of variable ageing of diesel emissions in western nations and showed that $E_{\text{Abs}}$ was generally below 1.5. However, subsequent measurements have found $E_{\text{Abs}}$ up to 2 in aged pollution in Beijing in both summer and winter (Xie et al., 2019a, b). Comparisons to Mie

models are complicated due to size-dependent underprediction of the MAC of bare soot at shorter visible wavelengths, size-dependent overestimation of $E_{\text{Abs}}$, and only a moderate level of constraint on the BC refractive index (Bond and Bergstrom, 2006).

The addition of coatings may also cause fractal soot aggregates to collapse into a quasi-spherical shape, and Li et al. (2003) observed compacted soot particles using transmission electron microscopy on samples of aged haze from biomass burning

in southern Africa. This could reduce absorption as less of the soot is exposed to the light (Scarnato et al., 2013), or increase absorption due to stronger interactions between neighbouring soot spherules (Liu et al., 2008). The relative importance of these two competing effects is wavelength-dependent in complex optical models, but experimental evidence of either effect remains sparse.

BrC is another major absorbing component of combustion aerosol, and this can include absorption by an emerging classifi-

cation type for BrC referred to as tar balls. Primary emissions of BrC are particularly prevalent in biomass burning smoke, and secondary BrC may form photochemically or by condensation as plumes cool. BrC absorbs strongly at shorter visible wavelengths, but has a higher AAE than BC (typically greater than 2), and is not a strong absorber at longer visible wavelengths (Lund Myhre and Nielsen, 2004). This means that AAE may be used to discriminate between the relative contributions of BrC and BC to total absorption. Primary BrC becomes less absorbing with time due to photochemical bleaching. The majority

of BrC absorption decays with a half-life in the region of 9 – 15 hr (Lee et al., 2014; Forrister et al., 2015), but laboratory studies have shown that some BrC species are particularly resistant to some bleaching pathways, such as reactions with ozone (Browne et al., 2019). Moreover, aged biomass burning aerosol may also have a significant BrC component arising from consistent mechanisms acting towards secondary BrC formation over the aerosol atmospheric lifetime.

In climate models, optical property schemes are chosen based on simplicity of implementation and computational efficiency.

Schemes based on Mie theory are relatively simple to implement, requiring only information on particle size and the relative fractions of different components. Optical models that explicitly include particle shape are too complex and computationally expensive to be used in this context, as well as having limited constraint from observations. In many cases, different aerosol components are mixed together to form homogeneous "'grey spheres"', each of which has a single complex refractive index that is often estimated through a simple linear volume-weighting mixing rule (e.g. Ghan and Zaveri, 2007; Bellouin et al.,

2013). Core/shell bin schemes have also been developed (e.g. Jacobson, 2001; Matsui et al., 2013), but widespread implementation is yet to take place as they are more computationally expensive than grey spheres, and it is yet to be demonstrated that they necessarily give more accurate results. Bond et al. (2006) discussed the limitations of these two types of scheme, and highlighted the potential of each to reach unphysical results under certain circumstances. Recently, several studies have produced parametrisations to calculate MAC or $E_{\text{Abs}}$ for variable internal mixing of BC with other aerosol components (Liu

et al., 2017; Chakrabarty and Heinson, 2018; Wu et al., 2018). These have been constructed using some type of empirical fit





to MAC or $E_{\text{Abs}}$, using real-world measurements and/or calculations from complex optical models. However, none of these schemes have been explicitly tested on highly aged biomass burning smoke in ambient conditions.

The southeast Atlantic is an excellent natural laboratory for studying the properties of aged smoke during the southern hemisphere biomass burning season, roughly May – October (Zuidema et al., 2016). At this time of year, the southeast Atlantic region is devoid of notable convective clouds that would contribute to removal of aerosol particles from the lofted smoke layers while they remain in the free troposphere. Combined with the high strength of solar radiation in the tropics, this allows the smoke to reach a very high photochemical age. Zuidema et al. (2018) recently reported high values of MAC measured using ground-based instruments on Ascension Island, where aged smoke had entrained into the boundary layer, with equivalent $E_{\text{Abs}}$ of 1.7 − 2.3. In this paper we describe results from the CLARIFY field campaign, which involved airborne measurements of black carbon and optical properties in highly aged biomass burning plumes over the southeast Atlantic (Haywood et al., 2020, in prep.). We quantify the range of values of measured MAC and AAE in this highly aged biomass burning smoke in both the boundary layer and free troposphere, and use single-particle measurements of BC mass and mixing state to assess the suitability of several optical models and parametrisations to simulate aerosol absorption in this environment.

## 2 Experimental

### 2.1 CLARIFY measurement campaign

The CLARIFY project took place between 16th August and 7th September 2017. The Facility for Airborne Atmospheric Measurements (FAAM) BAe-146 airborne research aircraft was based out of Ascension Island (7.97°S, 14.40°W). Smoke plumes from the African continent take roughly 4 − 8 days to travel from source to the measurement area (Adebiyi and Zuidema, 2016; Gordon et al., 2018). In total, 28 flights were performed, and the total flight duration over all flights amounted to 100 hours. Fig. 1 shows the aircraft tracks of all flights included in this analysis. All science flights took off and landed on Ascension Island. Full details of the rationale and implementation of the flight campaign are given by Haywood et al. (2020, in prep.).

### 2.2 Instrumentation

The FAAM aircraft was fitted with a suite of instrumentation for making online measurements of the physical and chemical properties of aerosols, cloud microphysics, remote sensing, and meteorological variables such as temperature, pressure, and relative humidity. Only the instrumention directly relevant to the measurements presented in this study are described here. Black carbon number and mass concentrations and single-particle mass and mixing state were measured using a single-particle soot photometer (SP2). The instrumental setup on the aircraft have been described previously (McMeeking et al., 2010). The SP2 measures the scattering cross-section of particles passing though its laser beam, and particles containing refractory black carbon (rBC) are heated to their incandesence temperature. The intensity of the incandescent light emission is proportional to the mass of rBC in the particle. The SP2 alignment was checked using nebulised 200 nm and 300 nm polystyrene latex spheres,





and these measurements were also used to calibrate the SP2's scattering channel. The instrument response to incandescent rBC was calibrated several times throughout the campaign using nebulised Aquadag, in the manner described by Laborde et al. (2012b). In biomass burning plumes the overall uncertainty of the BC mass concentration is $\pm17\%$ as a combination of 14% uncertainty due to the uncertainty in the calibration, and 9% variability between instruments (Laborde et al., 2012b).

Single-particle BC mass was converted to mass-equivalent BC diameter ($D_{\mathrm{BC}}$) using a BC density ($\rho_{\mathrm{BC}}$) of $1.8\,\mathrm{g\,cm^{-3}}$ (Bond and Bergstrom, 2006). We also used leading-edge only (LEO, Gao et al., 2007; Liu et al., 2015) fits to the SP2's scattering measurement to derive the scattering cross-section for BC particles at a wavelength of $1064\,\mathrm{nm}$, and we use a method similar to that described by Taylor et al. (2015) to quality-assure the LEO data. This method is discussed further in the results section. The BC scattering cross-section data can then be combined with an optical model to provide physical properties of the particle, such

as the spherical-equivalent shell/core ratio, or the mass ratio (MR) of non-BC to BC components. These optical calculations are discussed in more detail in Sect. 4.

   Non-refractory aerosol composition was measured using a compact time-of-flight aerosol mass spectrometer (AMS), which we used to provide mass concentrations of organic aerosol (OA), as well as the major inorganic species $SO_4$, $NO_3$, and $NH_4$. The AMS was calibrated before and after each flight using nebulised ammonium nitrate, and the relative ionisation efficiencies

of $NH_4$ and $SO_4$ were also calibrated during the campaign by varying the concentrations of nebulised ammonium nitrate and ammonium sulphate (Allan et al., 2004).

   Carbon monoxide (CO) concentrations were measured using a vacuum ultraviolet florescence spectroscopy monitor AL5002. The CO monitor was calibrated and zeroed multiples times per flight at different altitiudes, and has an overall accruacy of $\pm3\%$.

   Aerosol absorption coefficient ($B_{\mathrm{Abs}}$) was measured using three separate photoacoustic spectroscopy (PAS) cells, operating

at wavelengths of $405\,\mathrm{nm}$, $514\,\mathrm{nm}$, and $655\,\mathrm{nm}$, as part of the new EXtinction, SCattering and Absorption of Light for AirBorne Aerosol Research (EXSCALABAR) suite of instrumentation (Davies et al., 2019; Szpek et al., 2020, in prep.). The PAS instruments sampled behind an impactor with a cut-point of $1.3\,\mathrm{\mu m}$ aerodynamic diameter to remove coarse-mode particles, and a nafion drier to lower the measurement relative humidity (RH) to below 10% within the PAS detection volume. Davies et al. (2018) describe the PAS instrument and ozone calibrations, which were performed before each flight. For long

averaging times, the total uncertainty on the $B_{\mathrm{Abs}}$ measurement is $8\%$ (Davies et al., 2019).

   We calculate MAC by dividing $B_{\mathrm{Abs}}$ by the BC mass concentration measured by the SP2, and the systematic uncertainty in MAC of $\pm\,19\%$ is calculated by combining the relative uncertainties of $B_{\mathrm{Abs}}$ and BC mass concentration in quadrature. AAE was calculated by comparing the absorption coefficient at different wavelengths. The $8\%$ uncertainty in the absorption coefficient applies almost equally to all channels as it derives from a comparison between ozone calibrations and nebulised

nigrosin dye (Davies et al., 2018). The uncertainty in AAE is therefore significantly lower than that of individual single wavelength PAS measurements and is expected to be under 5%.

   All aerosol concentration measurements were corrected to standard temperature and pressure of $273.15\,\mathrm{K}$ and $1013.25\,\mathrm{hPa}$. The inboard aerosol instruments all sampled from Rosemount inlets, and the instrumental setup and detection ranges mean they all sampled a comparable size range of accumulation mode submicron aerosol. In-cloud data were removed when the liquid

water content measured by a wing-mounted cloud droplet probe exceeded $0.01\,\mathrm{g\,m^{-3}}$. To prevent noise at low concentrations





affecting our measurements, aerosol data were also removed when the BC mass concentration fell below $0.1\ \mu\mathrm{g\,m^{-3}}$ or the absorption coefficient at any wavelength fell below $5\ \mathrm{Mm^{-1}}$.

## 3  Results

Figure 2 shows vertical profiles describing the average atmospheric conditions measured during CLARIFY in terms of the level of pollution, meteorological variables, and chemical composition of the aerosol. The marine boundary layer (MBL) was capped by an inversion of several degrees Celsius around $1.5 - 2\ \mathrm{km}$ in altitude, though the exact altitude and depth of this inversion varied. The RH in the boundary layer increased from 75 % near the surface to to 80% up towards the top of the boundary layer, where a stratocumulus deck was often present, though excluded from our data. Plumes in the free troposphere were much drier but the RH increased with altitude, from 20% at $2\ \mathrm{km}$ to 50% at $5\ \mathrm{km}$, and some plumes reached up to $70 - 80\%$.

The CO and BC profiles show the altitudes at which the biomass burning smoke reached the area around Ascension Island. There was considerable variability in the regimes that were experienced during the aircraft deployment with the pollution confined either to the MBL or free troposphere, or pollution in both the MBL and free troposphere at any particular time (Wu et al., 2020). However, for the purposes of the analysis presented here, where we focus on aerosol intrinsic properties, we simply present average vertical profiles. The highest smoke concentrations were found between the inversion top and around $4.5\ \mathrm{km}$, and no significant concentrations of smoke were intercepted above $5\ \mathrm{km}$. The variability in the free troposphere CO and BC profiles shows the average of many discrete plumes at variable altitudes. The aged smoke particles were in the accumulation mode size range (Wu et al., 2020). Comparing the BC number concentration to the accumulation mode number concentration measured by a passive cavity aerosol spectrometer probe (PCASP), the BC number fraction was $39 \pm 7$ %. Using the SP2 alone, comparing to the SP2's scattering measurement (which has a more limited detection range), this fraction was $55 \pm 7$ %. The exact numbers are sensitive to the lower detection limits of the various measurements, and may therefore vary from probe to probe. Nevertheless, it is clear that a significant fraction of particles contained BC, but also that a significant fraction contained little to no BC.

Panels (e) – (h) in Fig 2 show the ratios of non-refractory aerosol components measured by the AMS, relative to the BC mass concentration, which give an indication of the composition of BC coatings if the particles are internally mixed, as well as the composition of particles that contain little to no BC. OA dominated the aerosol mass. Although the OA/BC ratio profile was relatively constant throughout the profile compared to ratios of some of the inorganic components to BC, it was larger and more variable in the MBL than the FT. In the free troposphere, the $NO_3/BC$, $NH_4/BC$ ratios all increased between $2 - 5\ \mathrm{km}$, as did the OA/BC ratio although to a much lesser extent. This is consistent with a change in the equilibrium of semi-volatile species to favour the condensed phase at lower temperatures, which is discussed in more detail by Wu et al. (2020). However, these differences may also be related to different ageing and/or sources between the smoke layers measured at different altitudes. The increased sulphate and ammonium in the MBL compared to the free troposphere suggests a marine influence to the aerosol,





and the variability of these components in the MBL is due to variation in the relative contributions of biomass burning smoke and marine aerosol.

Figure 3 shows average vertical profiles of BC properties and MAC. The average values of the BC core mass median diameter (MMD) are slightly smaller than most field measurements of biomass burning (e.g. Sahu et al., 2012; Taylor et al.,
2014), which tend to be in the range $190 - 215\,\mathrm{nm}$, but they are still larger than laboratory burns by May et al. (2014), which were in the range $170 \pm 20\,\mathrm{nm}$. Holder et al. (2016) also reported that the MMD of fresh BC from biomass burning was source dependant, showing that grass such as from savannah sources in southern and central Africa produced smaller BC cores. The count median diameter (CMD) values are also reported here, as they can be a useful indicator of different sources or be an indication of cloud processing. Literature comparisons of CMD are difficult, as the CMD is sensitive to the lower cut-off
diameter of the instrument, which varies between instruments (Laborde et al., 2012b). The vertical profiles show no strong trends in CMD and MMD, though both distributions were shifted a few nanometres lower in the boundary layer compared to the free troposphere. This shift could be due to different source regions, or wet removal processes in the boundary layer preferentially removing the largest particles, though to a much lesser extent than described by Taylor et al. (2014).

Figure 3 (c) shows the profile of median BC MR and the spherical-equivalent shell/core ratios, assuming a concentric
core/shell geometry of a BC core coated by non-BC material. The average values show significant internal mixing throughout the profile, and compare well to previous measurements of aged biomass burning smoke (e.g. Taylor et al., 2014, and references therein). The equivalent median absolute coating thicknesses were around $90\,\mathrm{nm}$ in the boundary layer, similar to aged smoke in Amazonia (Darbyshire et al., 2019), and up to $120\,\mathrm{nm}$ in the free troposphere. BC coatings increased with altitude; the median shell/core ratios were around 2.3 in the boundary layer, increasing up to around 2.6 between $4 - 5\,\mathrm{km}$. This trend
is likely to be related to the increase with altitude of condensed phase semi-volatiles such as ammonium nitrate in the free troposphere, as shown in Fig. 2.

Figure 3 (d) – (f) show the average vertical profiles of MAC at the three measurement wavelengths, and the campaign average values are also shown as a function of wavelength in Fig. 4. The average values were around a factor of 1.8 higher than that expected of fresh, externally mixed BC, reported by Bond and Bergstrom (2006). This high absorption enhancement
($E_{\mathrm{Abs-MAC}}$, the ratio of the measured MAC to the values reported by Bond and Bergstrom (2006)) is conceptually consistent with the high measured coating thicknesses, although the exact modelled values will depend on which optical model is used. The vertical variability in MAC was of the order of a few percent, and was smaller than the uncertainty of $\pm 19\%$ associated with the MAC calculation.

Alongside the mean MAC, Fig. 4 also shows the mean AAE calculated between the three wavelength pairs ($\mathrm{AAE}_{405-514}$,
$\mathrm{AAE}_{514-655}$, and $\mathrm{AAE}_{405-655}$ between $405 - 514\,\mathrm{nm}$, $514 - 655\,\mathrm{nm}$, and $405 - 655\,\mathrm{nm}$ respectively). The rationale behind examining the different wavelength pairs is that $\mathrm{AAE}_{405-514}$ is most sensitive to BrC absorption at the shorter wavelengths whereas $\mathrm{AAE}_{514-655}$ is less sensitive to BrC, and $\mathrm{AAE}_{405-655}$ is most useful for literature comparison. $\mathrm{AAE}_{514-655}$ was just below unity, which is within the range expected from absorption by black carbon, with no significant absorption by BrC at these wavelengths. An $\mathrm{AAE}_{405-514}$ of around 1.4 suggests a small contribution from BrC to absorption at the shorter wavelengths.
By extrapolating the BC absorption from the longer wavelengths, we estimate the fractional contribution of BrC to absorption





at 405 nm was $10 \pm 2\%$. It is likely that this BrC fraction of absorption will increase at even shorter wavelengths in the UV spectrum, although these are less relevant for climatic absorption due to the shape of the solar spectrum. The uncertainty here is extrapolated only from the relative uncertainties between the different PAS absorption wavelengths (<5% for the AAE values), as the BC mass concentration scales the MAC equally at all wavelengths, and drops out of the calculation. The error bars in

Fig. 4 are therefore not representative of the uncertainty in this calculation.

It is useful to calculate the effective MAC of the mixture of BrC and non-absorbing OA components, by dividing the calculated BrC absorption by the measured OA mass concentration, as this allows for comparison between different projects. This calculation yields the effective MAC for OA at 405 nm of $0.27 \pm 0.08 \ \mathrm{m^2 \, g^{-1}}$. Here the uncertainty is propagated from the relative uncertainties between the different PAS absorption wavelengths and the uncertainty in the OA mass concentration.

This value is not the MAC of BrC- it is the effective MAC of the mixture of BrC and nonabsorbing OA components. Literature comparison here is difficult as many studies use different wavelengths and different methods to quantify the OA or BrC mass. Two studies using a similar measurement method and wavelength as described here reported values of $0.5 - 1.5 \ \mathrm{m^2 \, g^{-1}}$ (Lack et al., 2012) and $0.53 - 0.6 \ \mathrm{m^2 \, g^{-1}}$ (Zhang et al., 2016) for BrC absorption fractions of around 30% at similar blue wavelengths. Our calculated effective OA MAC is significantly lower than these studies, which is consistent with the lower BrC absorption

fraction measured for this work.

## 4   Optical modelling

In the previous section we presented an overview of the vertical profiles and properties of aged smoke over the southeast Atlantic measured during the CLARIFY campaign. In this section, we describe the steps required to simulate MAC from the single-particle measurements of BC mass and scattering cross-section, and compare results from a variety of optical models to

the measured properties of the aerosol. The framework is built upon a combination of previous work by Taylor et al. (2015) and Liu et al. (2017), and is made up of two distinct halves. The previous work focused on relating the instrument response of the SP2 to physical properties of the particles. Taylor et al. (2015) detailed how to account for the limited detection range of the instrument, and also explored the sensitivities to assumptions about the BC density and refractive index ($m_{\mathrm{BC}}$). Liu et al. (2017) then compared optical models to extensive sets of lab and field measurements to produce a semi-empirical relationship

relating BC particles' optical scattering to size and mixing state, essentially characterising how real particles deviate from the core/shell Mie model. Applying these steps generates a set of physical properties of the BC-containing particles, similar to what may exist within a climate model. For the first half, previous work has determined a well-characterised route to determine the size and mixing state of the BC particles with a degree of accuracy (Liu et al., 2017).

These physical properties are then used to test different optical schemes that could be implemented in such a climate model,

given the in situ optical properties observations of MAC and AAE above. Here we have freedom to vary the optical model and underlying assumptions, including parameters such as $m_{\mathrm{BC}}$, as would be possible in a climate model. The optical models and parametrisations tested in this analysis are listed in Table 1, and described in detail in supplementary Sect. S1. We have tested the core/shell Mie model, as well as several homogeneous grey sphere models, which utilize a Mie model with a sphere





of one single complex refractive index, calculated using different rules to account for the mixing between BC and non-BC components. We have also tested several parametrisations of either MAC or $E_{\text{Abs}}$ (Liu et al., 2017; Chakrabarty and Heinson, 2018; Wu et al., 2018).

### 4.1 Deriving a 2-D size and mixing state distribution

The SP2 measures the per-particle BC mass ($M_{\text{BC}}$) and scattering cross-section over a specified collection angle. In studies of BC mixing state, a lookup table is often used to estimate some metric of internal mixing, such as the coated diameter assuming a spherical core/shell morphology. This process is summarised in Figure 2 of Taylor et al. (2015) and associated discussion. Some recent work has focused on the mass ratio (MR) as an alternative metric, defined here as MR$= M_{\text{non-BC}}/M_{\text{BC}}$ (Liu et al., 2017). The advantage of using MR as a metric is that it does not assume anything about particle morphology. The disadvantage

is that explicit optical models do not use MR. Commonly-used Mie models work in terms of diameter (or more specifically, the size parameter $\chi = \pi D/\lambda$, where $\lambda$ is the wavelength), so some measurement or assumption of the densities of both the core and coating are needed to convert MR into diameter. The process for generating a 2-D mixing state distribution takes several steps:

1. Perform core/shell Mie scattering calculations at 1064 nm to create a 2-D lookup table of scattering cross-section versus
core diameter and coated diameter

2. Correct this Mie table to represent light scattering by real BC particles using the empirical correction described by Liu et al. (2017), by calculating the equivalent MR using the diameters and densities of the core and coating

3. Convert the single-particle $M_{\text{BC}}$ measured by the SP2 to the spherical-equivalent $D_{\text{BC}}$, giving single-particle data of $D_{\text{BC}}$ and scattering cross-section

4. Process single-particle data through the table to give single-particle spherical-equivalent core and shell diameters, including the Liu et al. (2017) correction

5. Convert the single-particle data to equivalent $M_{\text{BC}}$ and MR, and bin the data into a 2-D distribution of MR vs $M_{\text{BC}}$

6. Correct this distribution for the limited detection range of the SP2

These steps require knowledge of several intrinsic properties of the particles. A BC density of $\rho_{\text{BC}} = 1.8\,\text{g\,cm}^{-3}$ is generally
accepted as the best estimate after the review by Bond and Bergstrom (2006). Previous work has shown that using the Mie core/shell model with this density and a BC refractive index $\boldsymbol{m}_{\text{BC}} = (2.26 - 1.26i)$ produces good agreement for externally mixed BC particles for scattering at 1064 nm (Moteki et al., 2010; Taylor et al., 2015). These values were also used to derive the empirical correction by Liu et al. (2017). For the refractive index of the shell's non-BC components, we used a value of $1.5 - 0i$ as in previous work (e.g. Schwarz et al., 2008; Taylor et al., 2015; Liu et al., 2017). Using a non-absorbing coating assumes
that BrC makes no direct contribution to absorption. The impact of this assumption will be discussed in the next section. The density of non-BC components was calculated by volume mixing using the relative fractions of the AMS composition,





applying densities of 1.2 g cm$^{-3}$ for OA, 1.77 g cm$^{-3}$ for inorganic components (Cross et al., 2007), assuming there is no chemical difference between the BC coating and the bulk non-BC mass of the aerosol.

Figure 5 shows a 2D distribution of BC mass and mixing state (MR versus $M_{BC}$), which was generated by taking the SP2 measurements and applying the semi-empirical Liu-$E_{Abs/Sca}$ scheme (steps 4 & 5 above). The MR distribution was centred

around 20 for 1 fg cores, but was around 7 – 8 for BC larger than a few femtograms. Few particles were in the bottom bin, of MR = 0 for externally mixed particles. The distribution has been corrected for the size-dependent detection efficiency of the SP2 instrument (step 6 above), using the approach described by Taylor et al. (2015), and discussed in more detail in Sect. S2. Previous studies have used a core/shell Mie model to relate the light scattering properties of particles to their coating properties. The Liu-$E_{Abs/Sca}$ is an empirical correction to the core/shell Mie model that accounts for the fact that particles with MR < 3 do

not scatter light exactly as described by the core/shell Mie model at 1064 nm. In our dataset the vast majority of particles had MR >3, and therefore were in the regime where the Liu-$E_{Abs/Sca}$ parametrisation shows that light scattering at 1064 nm is well represented by the core/shell Mie model.

Taylor et al. (2015) investigated the effect of uncertainties associated with assumed parameters on the derived coating thicknesses. Here we performed a Monte Carlo analysis to calculate the combined uncertainty in the derived coating properties

and MR. A description of the Monte Carlo approach, as well as a summary of the results, is presented in Sect. S3. While the uncertainties in the shell/core ratios and absolute coating thicknesses were around 6 – 8%, the uncertainty in the average MR was around 20%. This larger uncertainty is likely due to the larger uncertainty in the density of the coating material. In the next section we use the 2-D distributions of MR versus $M_{BC}$ as input for various optical models to generate absorption properties, and discuss the uncertainty associated with these calculations.

### 20  4.1.1  MAC and AAE calculations

Having established the properties of the measured particles in terms of $M_{BC}$ and MR, it is then possible to use the different optical models in Table 1 to calculate the MAC of the BC particles. The overall process is in some ways conceptually similar to the last section, but starts with the 2-D distribution of MR vs $M_{BC}$, rather than the single particle data, and outputs the ensemble mean absorption coefficient. The different optical models described in Sect. S1 were used to calculate tables of

absorption cross-section on the same grid as the 2-D particle distributions, converting to spherical-equivalent diameters where necessary using the same densities as in the previous section. The uncertainties associated with the MAC and AAE calculations from the different optical models were calculated using the Monte Carlo analysis described in Sect. S3.

For calculating absorption at visible wavelengths, various values of $m_{BC}$ were tested in the Mie models, and these are listed in Table S1. For the parametrisations we used the values used in their derivation. The rationale behind varying $m_{BC}$ in this way

is simply to explore the sensitivity to this parameter. There is no consensus on the best value of $m_{BC}$ (or if there is one fixed value), and different studies use different values, often with little explanation as to why a particular value has been chosen. Bulk measurements of visible absorption suggest values in the range specified by Bond and Bergstrom (2006), and these values are also used in some climate models (e.g. Conley et al., 2012). The aim of our study is to investigate the performance of different models within the framework of how they are currently implemented or could be implemented in the future, rather than to use





our results to determine the best value of $m_{BC}$. Since these particles are unlikely to be perfectly spherical, any Mie model of their optical properties is an approximation, so it is feasible that absorption and scattering could vary from model predictions in different ways at different wavelengths.

For each of the three measurement wavelengths, Figs. 6 and 7 show the average values of MAC and AAE for 71 straight and level runs measured during CLARIFY at altitudes between 50 – 5700 m, calculated using the various aforementioned optical models, alongside the average of the ambient observations. The horizontal axes in Fig. 6 are $k_{BC}$, the imaginary component of $m_{BC}$, but it is important to recognise that the real part of the refractive index was varied simultaneously according to Table S1. Run-to-run variability in the modelled and measured values was of the order a few percent, so comparing the campaign average values is appropriate here.

Modelled MAC values varied significantly between models as well as different assumed $m_{BC}$. The grey sphere models (the dashed lines in Figs. 6 and 7) universally produced values of MAC that were higher than the measured values, other than for the smallest values of $m_{BC}$ at the blue wavelength. At the green and red wavelengths the modelled MAC was around 20% higher than the measurements for the smallest values of $m_{BC}$, and up to 100% higher for the largest values of $m_{BC}$. Meanwhile, at the blue wavelength, the grey sphere models overestimated the MAC by 7 – 60% (dependent on mBC), although this overestimation increased to values in the range 18 – 77% once the effect of BrC had been removed from the MAC estimation. The $AAE_{514-655}$ values from the grey sphere models agreed well with the measurements for the lowest values of $m_{BC}$, but were too low for the highest values. $AAE_{405-514}$ was well below unity for all values of $m_{BC}$ for all the grey sphere models, and the values produced were outside the range expected for BC even in the absence of BrC (i.e. close to one).

The core/shell Mie model consistently underestimated MAC at the blue wavelength, but was within the uncertainty of the measurements at the green and red wavelengths for most values of $m_{BC}$. The blue MAC was 16 – 36% lower than the measured values depending on the value of $m_{BC}$ whereas the green MAC ranged from 21% below to 13% above the measurements, and the red MAC from 16% below to 29% above the measurements. When the calculated BrC absorption was subtracted, the core/shell Mie values for the blue MAC were still up to 29% lower than the measurements. For most values of $m_{BC}$, the core/shell Mie model calculations were within the experimental experimental uncertainty of the measured MAC at green and red wavelengths.

The wavelength dependence to the relationship between core/shell model MACs and measured values manifests as low values of AAE values, as shown in Fig. 7. The underprediction of MAC at short wavelengths in the core/shell model is explained in supplementary Sect. S4. In short, the underprediction by core/shell Mie theory arises because the mixing state and morphology of the measured aerosol is different to that assumed in the derivation of Mie theory. In the Mie and core/shell Mie models, the skin depth effect prevents light interacting fully with all the light absorbing BC. The reality is that the BC is likely a nonspherical aggregate with a high surface-to-volume ratio, and this high surface area relative to the total BC mass allows light to interact fully with the BC component and the skin depth effect is negligible. Therefore, the skin depth effect causes Mie models to under-predict the light absorption properties for the BC aerosols under investigation here. See Figure 4(d) vs Figure 4(f) in Chakrabarty and Heinson (2018) for the model calculations that demonstrate this interplay between shielding and the fractal surface-to-volume ratio.





To explicitly demonstrate the effect of the skin-depth shielding, we calculated the MAC based on the $E_{\mathrm{Abs}}$ from the core/shell Mie model multiplied by the baseline BC MAC from Bond and Bergstrom (2006), and termed this calculation CS-$E_{\mathrm{Abs}}$. $E_{\mathrm{Abs}}$ here refers to the ratio of coated MAC to uncoated MAC within the Mie model, rather than $E_{\mathrm{Abs\text{-}MAC}}$ which is the ratio of coated MAC to the Bond and Bergstrom (2006) values. In contrast to the core/shell Mie model results, the use of CS-$E_{\mathrm{Abs}}$ gave

MAC values that agreed with the measurements within 18% at all wavelengths and $m_{\mathrm{BC}}$, which was within the experimental uncertainties, whether or not the calculated BrC absorption had been subtracted. The MAC from CS-$E_{\mathrm{Abs}}$ was dependent on $m_{\mathrm{BC}}$, but the range of $m_{\mathrm{BC}}$ tested in Fig 6 gave only up to 15% variation in MAC, which was much less sensitive than the other Mie models. The AAE values using CS-$E_{\mathrm{Abs}}$ were $0.91 - 0.97$, with a very weak dependence on $m_{\mathrm{BC}}$. This range agrees well with the aircraft measurements of $\mathrm{AAE}_{514-655}$, but was well below the measurements of $\mathrm{AAE}_{405-514}$, as expected since

the effect of BrC is not accounted for in the model.

The parametrisations shown in Fig. 6 (d) – (f) were generally more successful than the pure Mie models, and the Chak-$E_{\mathrm{Abs}}$, Chak-MAC, and Liu-$E_{\mathrm{Abs/Sca}}$ schemes all gave results within the uncertainty range of the measured MAC. The Chak-MAC parametrisation values of MAC agreed within 6% of the measurements at all wavelengths, and Chak-$E_{\mathrm{Abs}}$ predicted MAC values $18 - 21$% larger than the measured values when the calculated BrC absorption was removed at 405 nm, or $9 - 18$% if

not. The results from Liu-$E_{\mathrm{Abs/Sca}}$ agreed within 2% at all wavelengths when the calculated BrC absorption was removed, or 9% if not. The MAC calculations using the Wu-$E_{\mathrm{Abs}}$ scheme were around 25% below the measurements at all wavelengths, and there was only a small overlap in the uncertainty ranges between the model and measured values. In terms of AAE, the Liu-$E_{\mathrm{Abs/Sca}}$ and Wu-$E_{\mathrm{Abs}}$ parametrisations produced values in excellent agreement with the measured $\mathrm{AAE}_{514-655}$, and the Chak-MAC and Chak-$E_{\mathrm{Abs}}$ schemes both have AAE fixed at exactly 1, which was also close to the measured value. All

parametrisations underestimated $\mathrm{AAE}_{405-514}$ by some margin, which is expected as they are strictly applicable to black carbon aerosols only and do not account for absorption by BrC.

## 5   Discussion

### 5.1   Physical and optical properties of highly aged biomass burning soot

The main focus of this paper has been the measurement of the physical and optical properties of black carbon in heavily aged

biomass burning plumes, and a comparison of the ability of different optical schemes to utilise these physical properties to predict the optical properties. One of the key questions in the field of aerosol absorption is the relative importance of black and brown carbon. This is particularly important in biomass burning plumes, where BrC is thought to have a strong effect near source that diminishes with age (Forrister et al., 2015).

Our measurements have shown that the $\mathrm{AAE}_{405-655}$ values in smoke plumes over the southeast Atlantic were always close

to 1 (with a mean $\mathrm{AAE}_{405-655}$ of 1.16), which was consistent between our airborne measurements in 2017 and ground-based measurements taken on Ascension the previous year over a similar wavelength range (Zuidema et al., 2018). Previous measurements of savannah fires in southern Africa showed AAE values across the whole visible wavelength range of $\approx 1.8$ in relatively fresh plumes, but this was reduced to 1.2 in regional haze layers (Kirchstetter et al., 2004). Our measurements are



consistent with this picture of aged haze having a reduced BrC contribution compared to freshly emitted aerosol. Other recent studies have also shown AAE close to 1 in African biomass burning smoke that had been aged for several days (Zuidema et al., 2018; Denjean et al., 2019). These optical properties are also in a similar range to measurements by Saturno et al. (2018), who measured smoke from southern Africa once it had reached Amazonia after 10 days transport. Their measurements found

average AAE of $\approx 0.9$, and a MAC of 12.3 at 637 nm, equivalent to $E_{\text{Abs-MAC}}$ of 1.9. One of the key features of these African smoke plumes is their long lifetime, and after several days ageing in the tropical sun, the impact of BrC on visible absorption is minor compared to BC.

From a modelling perspective, the BC number fraction is important as it has a large impact on the amount of material assumed to be internally mixed with BC. For example, in the modal aerosol scheme described by Bellouin et al. (2013), as

smoke ages it moves almost entirely into what they refer to as a "soluble accumulation mode". Consequently, not only do ~100% of particles contain BC, but all non-BC accumulation mode aerosol is internally mixed with that BC, and this has an important impact on optical properties (Matsui et al., 2018). The exact values for the BC number fraction are difficult to determine due to the different detection limits of the different probes, but the values here (~40% from SP2/PCASP and ~55% from the SP2 alone) are high compared to our previous measurements of 10% from the SP2 alone in boreal smoke

plumes (Taylor et al., 2014), but less than measurements in fresh diesel emissions or flaming plumes which reach 100%. It is important to note that in these aged plumes, the majority of particles by number contain little to no BC, and to ensure that this is replicated in any mixing state scheme used in climate models. The fraction of BC-containing particles is variable in different environments and this is important for calculating the optical properties. However, this number is not widely reported in the literature since it is operationally defined by the measurements and varies from experiment to experiment. We recommend that

in future, effort is given to constraining this number in a range of different environments.

The BC particles measured during CLARIFY were universally thickly coated. The concept of "coatings", and the use of optical techniques to measure them, is not always appropriate to describe internal mixing of BC with other material. Recent evidence from laboratory studies and multiple field campaigns (Liu et al., 2017; Pei et al., 2018) shows that for low values of MR, internal mixing with non-BC material takes the form of filling in voids between the soot spherules, and the process of

encapsulation only begins for higher values of MR, when the voids have already been filled. Liu et al. (2017) determined that the threshold value of MR, above which particles begin to behave optically in a way that resembles the core/shell model (in terms of $E_{\text{Abs}}$ and $E_{\text{Sca}}$), is around 3. Our median MR values were in the range $8 - 12$, which is in the range where Liu et al. (2017) showed that particles behave like core/shell calculations in terms of light scattering at a wavelength of 1064 nm. The concept of BC coatings therefore does seem to be appropriate for particles with values of MR this high, although it does not

necessarily mean that the core/shell Mie model is capable of accurately predicting any or all aspects of optical properties at visible wavelengths. These values of MR, and equivalent shell/core ratios, are higher than most measurements of fossil fuel BC (summarised by Pan et al. (2017)), but similar to some previous measurements of aged biomass and solid fuel burning emissions (e.g. Liu et al., 2014; Taylor et al., 2014), as well as aged pollution in several different regions of China (Gong et al., 2016; Xu et al., 2018; Zhao et al., 2020), and they fall short of the shell/core ratios >3 reported in the stratosphere by Ditas

et al. (2018).





Xu et al. (2018) provide a useful summary of previous measurements of $E_{\text{Abs}}$, based on the MAC comparison method used here, thermodenuder comparisons, and an aerosol filtration–dissolution method. Measurements towards the green/red end of the visible spectrum (where BrC has minimal absorption) seem to show a dichotomy between environments where coatings on BC cause a considerable lensing effect, and those where either the BC is largely externally mixed, or the limited internal mixing does not cause a significant lensing effect. Ambient measurements with a strong lensing effect (i.e. with long-wavelength $E_{\text{Abs}}$ around 1.5 or greater) tend to be from sites measuring high levels of highly aged pollution (days rather than hours), and particularly in highly polluted environments in Asia; Xu et al. (2018) list several examples from literature, as well as primary measurements of their own, and other more recent studies have also found similar values of $E_{\text{Abs}}$ in Beijing (Xie et al., 2019a, b). Our measurements show that highly aged biomass burning soot from southern Africa universally falls into this category, as we found $E_{\text{Abs}}$ values of around 2 that were invariant with wavelength, indicating that the absorption enhancement is caused by a lensing effect.

One possible alternative to a lensing effect would be that the MAC of BC itself is this high. However, these values of MAC are well in excess of the range of MAC for externally mixed BC reported in literature (Bond and Bergstrom, 2006), so this explanation is not plausible. Another possibility is that the absorption is significantly affected by particles that are essentially dark brown spherical balls that may be closely related to tar balls, so-called "intermediate absorbers" (IA) described by Adler et al. (2019). By comparing untreated and thermodenuded absorption measurements, the authors demonstrated laboratory measurements of IA with an AAE of 2.4 between 401 and 661 nm, and argued that the presence of IA particles explains their field measurements of an apparent $E_{\text{Abs}}$ of 3 at 661 nm. Let us consider a theoretical external mixture of coated BC with AAE of 1, and IA with AAE of 2.4. If, for example, the $E_{\text{Abs}}$ of BC at 655 nm was actually 1.5, and the other 25% of red absorption came from IA, this would propagate to 40% of absorption at 405 nm, and the resultant total $\text{AAE}_{405-655}$ would be 1.45. The apparent blue $E_{\text{Abs-MAC}}$ would then be 2.5. This situation is therefore not consistent with our observations, as our measured $E_{\text{Abs-MAC}}$ was 2.1 at 405 nm, and our measured $\text{AAE}_{405-655}$ from Fig. 4 was 1.16.

## 5.2 Optical modelling of highly aged biomass burning soot

The Mie models had mixed success in replicating the measured optical properties. The volume mixing scheme produced values that were too high at long wavelengths for all values of $m_{\text{BC}}$. The Bruggeman and Maxwell-Garnett mixing schemes produced results that were almost identical, and only narrowly fell within the uncertainty of the measurements for the lowest $m_{\text{BC}}$ index tested, $1.75 - 0.63i$, and only at 405 nm when the BrC absorption had been subtracted. However, Bond and Bergstrom (2006) demonstrated that using such a low value of $m_{\text{BC}}$ externally-mixed BC produced MAC that was too low. For uncoated BC, the skin-depth shielding effect would also become more significant, manifesting as low values of AAE. To produce agreement for AAE, $m_{\text{BC}}$ would have to be lowered to physically unrealistic values that fall outside the range of previous measurements.

For the core/shell Mie model, there were several values of $m_{\text{BC}}$ that gave MAC at the red and green wavelengths that agreed within the measurement uncertainties, but only the highest values of $m_{\text{BC}}$ generated MAC that fell within the measurement uncertainties at 405 nm, even when the BrC absorption had been subtracted. The wavelength dependence to this level of agreement is primarily due to the stronger skin-depth shielding effect at this shorter wavelength. We explore the sensitivity of



MAC to $m_{BC}$ in an illustrative example in Fig. S5, using a representative BC size distribution and coating thickness. In fact, within our illustrative example there are no values of $m_{BC}$ that produce MAC at all three wavelengths that agree with our observations when used in the core/shell model. Absorption initially increases with $m_{BC}$, but skin depth shielding limits the maximum value of MAC to a value lower than the measurements at 405 nm. Figure S5 also shows that while higher values

of $m_{BC}$ give higher values of MAC, up to a point, they also give lower values of AAE. For all values of $m_{BC}$ that fall in the region of measurements from literature, the AAE from core/shell Mie theory was well below the range of measured values in CLARIFY. One way around this could be to use a wavelength-dependent $m_{BC}$, tuned to give realistic values of AAE, but this would still struggle to be able to produce high enough MAC at 405 nm. For particles of these sizes, there are no values of $m_{BC}$ that can generate agreement with measured MAC and AAE within the framework of the core/shell Mie model.

The semi-empirical parametrisations were generally more successful at replicating both MAC and $AAE_{514-655}$. These schemes generated $AAE_{514-655}$ close to 1 as they either have AAE fixed at exactly 1 (Chak-MAC), or they use $E_{Abs}$ multiplied by the MAC from Bond and Bergstrom (2006), which is fixed at exactly 1. As $E_{Abs}$ has no strong wavelength dependence, all parametrisations gave values of AAE close to 1, and the Liu-$E_{Abs/Sca}$ and Wu-$E_{Abs}$ schemes gave values just below 1, that agreed with the measured values. All schemes generated $AAE_{405-514}$ well below the measured values, primarily because they

do not include the effects of BrC in the calculations. Comparing to literature, they are well within the range typically associated with BC absorption, and outside the range commonly measured for freshly emitted aerosol containing a high level of BrC (e.g. the AAE values of 2.5 measured by Lack and Langridge, 2013).

While we calculated a number for the effective MAC of bulk OA (including BC coatings and particles containing no detectable amount of BC, as a mixture of BrC and nonabsorbing OA), we have stopped short of determining a value for the

imaginary component of the OA refractive index. The MAC of OA can be calculated purely from the absorption and concentration measurements, whereas the refractive index determination requires detailed optical modelling. The implication of such modelling would be that every data point in Figs. 6 and 7 would give a different number. These numbers could only be accurately used in conjunction with the exact assumed BC optical properties associated with the relevant data point. This argument was made by Liu et al. (2015), and it is still valid in the context of this study. For some of the models, the BrC absorption would

have to be negative to agree with the MAC measurements. We would have no confidence in the reliability of any estimates of the imaginary component of OA refractive index generated in this way.

In terms of MAC, all the parametrisations gave values that agreed within the experimental uncertainties of the measurements, other than Wu-$E_{Abs}$. The best agreement was with the Chak-MAC and Liu-$E_{Abs/Sca}$ schemes, which came within a few percent of the measurements at all three wavelengths. It is not immediately clear why there is a difference between the Chak-$E_{Abs}$ and

Wu-$E_{Abs}$ schemes, as they are conceptually similar in the way they were generated. We speculate that it may be related to some detail of the morphology of the particles used in their simulations.

The apparent RI dependence of the $E_{Abs}$ parametrisations in Fig. 6 is potentially misleading. The CS-$E_{Abs}$ points show that $E_{Abs}$ is not strongly dependent on $m_{BC}$, meaning this apparent trend is not strongly related to $m_{BC}$. For values of $m_{BC}$ in the same region of the horizontal axis in Fig. 6, the CS-$E_{Abs}$ and Liu-$E_{Abs/Sca}$ schemes came out with almost identical values of

MAC. In this environment, with thick BC coatings and high values of MR, these two schemes are essentially the same.





We urge caution however; in low-MR environments such as fossil fuel emissions close to source, the core/shell model is likely to overestimate $E_{\text{Abs}}$ as it assumes all non-BC material takes the form of a coating, whereas evidence shows that for low values of MR this non-BC material is more likely to fill internal voids in the soot structure (Pei et al., 2018), which causes minimal absorption enhancement. The Liu-$E_{\text{Abs/Sca}}$ scheme is an empirical fit to correct for this overestimation of $E_{\text{Abs}}$, and the work by Chakrabarty and Heinson (2018) also included fits to particles with low MR, so these should give better results than the core/shell model in a low-MR environment.

Given the known inaccuracies in the Mie models when calculating the optical properties of atmospheric soot, and the success of some of the parametrisations in replicating the absorption properties of highly aged biomass burning aerosol, we recommend future work should investigate how to implement these types of schemes in a climate model. However, the parametrisations we have used remain incomplete for this purpose. A significant fraction, sometimes the majority, of visible light scattering in aged biomass burning plumes comes from particles containing black carbon. The scattering enhancement from BC coatings is an order of magnitude larger than the absorption enhancement, particularly at high humidities (Wu et al., 2018). Liu et al. (2017) considered $E_{\text{Sca}}$ at 1064 nm, but visible scattering for these size particles is more sensitive to shape than at 1064 nm. Wu et al. (2018) performed scattering calculations at 532 nm, but only produced a parametrisation for absorption. The ways in which scattering and absorption deviate from Mie calculations are not necessarily the same, and both should be investigated with equal importance if the end goal is to calculate variables like single scattering albedo for radiative forcing calculations.

## 6 Conclusions

We have presented a series of measurements of aerosol optical properties in southern African biomass burning smoke taken over the southeast Atlantic Ocean 4 – 8 days after emission, and a detailed investigation of the ability of different optical models to replicate these optical properties. Our dataset and analysis are unique in that they are the first set of measurements using high quality absorption and black carbon instruments, and the most detailed investigation of aerosol optical properties, in this type of environment. Our measurements also describe some of the thickest BC coatings, highest absorption enhancements, and most aged smoke plumes ever studied in ambient conditions. Smoke plumes that remain in the free troposphere over the southeast Atlantic have no deposition process, so they have a long lifetime as well as affecting the climate over a vast area thousands of kilometres wide. The high degree of ageing also represents an important real-world test of absorption calculations, which are normally based either on modelled particles, or observations much closer to source or in the laboratory.

Plumes were intercepted in the boundary layer and free troposphere up to altitudes of around 5 km, and the smoke was relatively homogenous in composition and optical properties. Based on measurements of the wavelength dependence of absorption, we estimate that BrC contributed around 10% of absorption at 405 nm, with an effective OA MAC of $0.27 \pm 0.08 \, \text{m}^2 \, \text{g}^{-1}$, but there was no BrC absorption at 514 nm and 655 nm. Absorption was dominated by black carbon, and thick coatings caused a wavelength-independent absorption enhancement of a factor of 1.8 compared to uncoated BC.

Mie models were able to successfully replicate some aspects of BC absorption, but failed at others, and absorption in all models was strongly sensitive to the assumed refractive index of the BC. To agree with the measured MAC and AAE, homoge-





neous grey sphere models would be required to use an unrealistically low value of the BC refractive index. The core/shell model produced MAC in the right range at the green/red end of the visible spectrum, but was unable to produce high enough values at blue wavelengths for any value of $m_{BC}$, and underestimated AAE by a significant margin. Some previous studies have shown that the core/shell model overestimates absorption enhancement for soot particles with lower levels of non-refractory material,

where coating is not an accurate description of the mixing state. Instead, our studies on thickly-coated particles demonstrate that the core/shell Mie model underestimates the MAC of the underlying BC at short wavelengths, with this underestimation in MAC caused by the optical skin depth of BC preventing light interaction with the total BC mass in a core/shell model that would otherwise occur in the true agglomerate structure of soot. There is no guarantee that the best implementation of a Mie model for our dataset would be the same for different environments, such as urban measurements closer to source. We

recommend any Mie model must be used with caution, if at all, when calculating aerosol absorption.

Two of the four semi-empirical absorption parametrisations we tested produced results that agreed with our observations for MAC within a few percent at all wavelengths, and three within the experimental uncertainties of the field measurements. Two parametrisations produced AAE values in the range of the aircraft measurements at the green/red end of the visible spectrum, but all underestimated AAE in the blue/green range as they did not include the contribution of BrC. Almost all the

Mie calculations gave values below the range of measured AAE. This work has been unique in testing the validity of these parametrisations for highly aged, thickly coated particles using real-world observations, and we have shown a good level of success for these schemes. Similar parametrisations should be developed for light scattering by internally mixed BC particles, and together these schemes may then be used to improve estimates of the aerosol direct and semi-direct effects.

*Data availability.* Airborne measurements are available from the Centre for Environmental Data Analysis https://catalogue.ceda.ac.uk/uuid/

38ab7089781a4560b067dd6c20af3769.

*Author contributions.* JWT prepared the manuscript and performed the bulk of the data analysis, with input from all coauthors. JWT, KB, IC, MF, PIW, JD, JML, MIC, CF, NWD, JMH and HC carried out the airborne measurements. JWT, HW, KS, IC, and JML processed the aircraft data. JML, MIC and HC advised on the data analysis. JMH, JML and HC are lead PIs who led the funding application and directed the research.

*Competing interests.* The authors declare no competing interests.

*Acknowledgements.* We thank everyone involved in the planning and execution of the CLARIFY project, as well as our hosts on Ascension Island. The BAe-146-301 Atmospheric Research Aircraft was flown by Airtask and managed by the Facility for Airborne Atmospheric





Measurements (FAAM), which is a joint entity of the Natural Environment Research Council (NERC) and the Met Office. CLARIFY was supported by NERC under grants NE/L013584/1 and NE/L013797/1.



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





**Table 1.** List of the different optical models used in this analysis. The absorption calculation column refers to whether the models/parametrisations calculate MAC or $E_{Abs}$. For those that only calculate $E_{Abs}$, MAC is calculated by multiplying $E_{Abs}$ by the fresh MAC value from (Bond and Bergstrom, 2006).

|  | Absorption calculation | Reference |
|---|---|---|
| **Coated sphere** | | |
| Mie core/shell | MAC | (Taylor et al., 2015) |
| CS-$E_{Abs}$ | $E_{Abs}$ | This study |
| **Homogenous grey sphere models** | | |
| Bruggeman | MAC | (Markel, 2016) |
| Maxwell-Garnett | MAC | (Bohren and Huffman, 1983) |
| Volume mixing | MAC | (Bohren and Huffman, 1983) |
| **Semi-empirical parametrisations** | | |
| Chak-$E_{Abs}$ | $E_{Abs}$ | (Chakrabarty and Heinson, 2018) |
| Chak-MAC | MAC | (Chakrabarty and Heinson, 2018) |
| Liu-$E_{Abs/Sca}$ | $E_{Abs}$ | (Liu et al., 2017) |
| Wu-$E_{Abs}$ | $E_{Abs}$ | (Wu et al., 2018) |



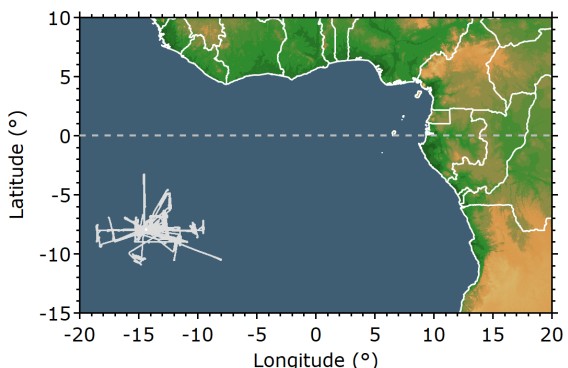

**Figure 1.** Map showing the location of the CLARIFY aircraft measurements included in this analysis (solid grey trace).



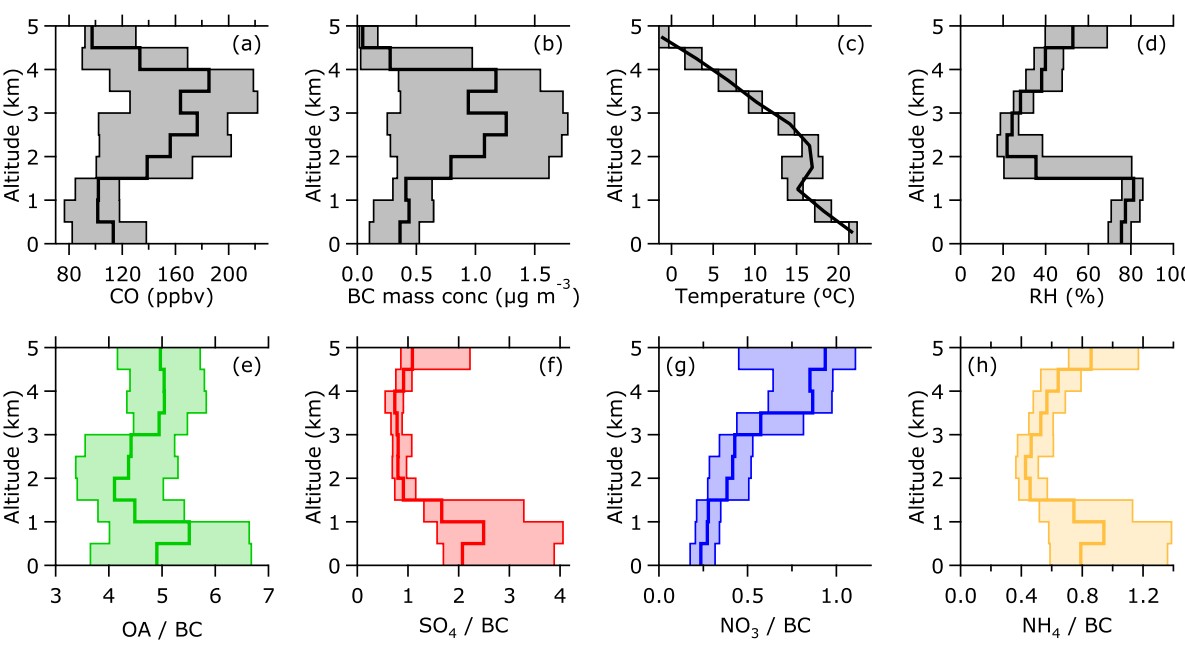

**Figure 2.** Campaign average vertical profiles of pollution levels, thermodynamic variables, and aerosol chemical composition. The solid lines show the median and the shaded areas show the 25th and 75th percentiles. Panels (c) – (h) show the in-plume data only. Panels (e) – (h) show the ratios of the mass concentrations of organic aerosol and the major inorganic ions to the mass concentration of BC.

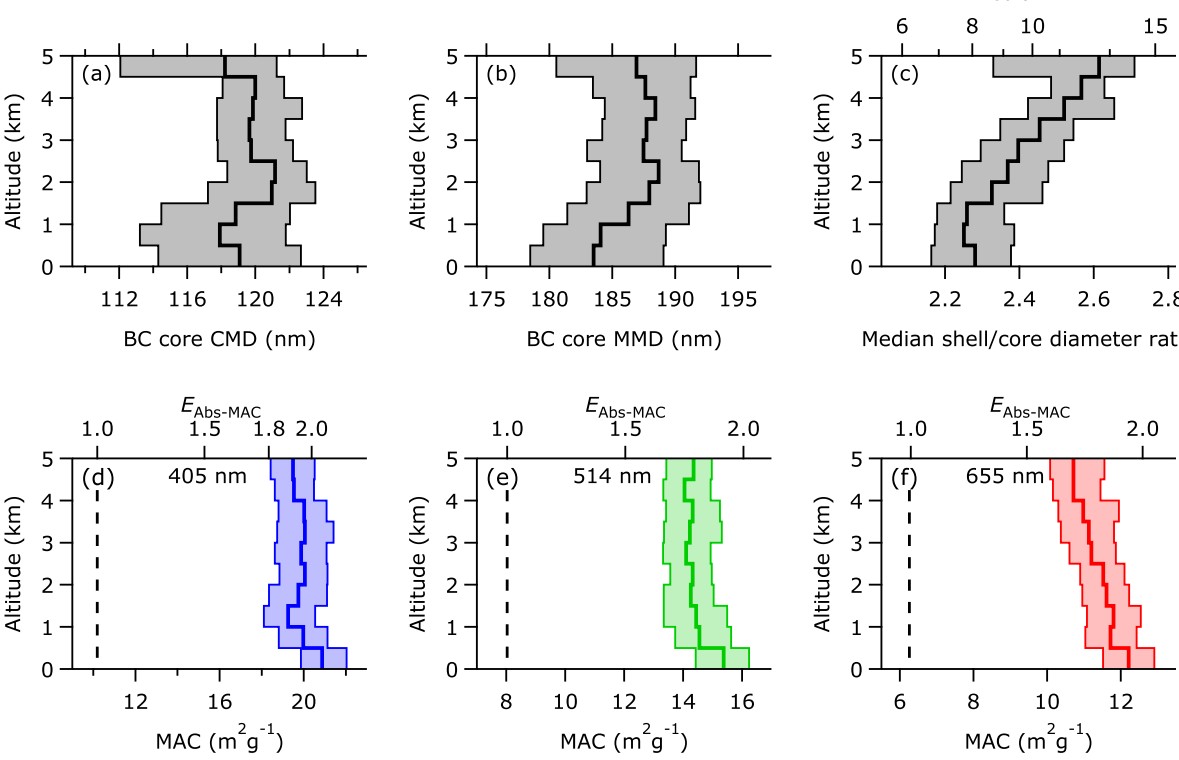

**Figure 3.** Campaign average vertical profiles of BC properties and MAC. The solid lines show the median and the shaded areas show the 25th and 75th percentiles. Panels (b) and (c) show the count median diameter and mass median diameter of BC core size distributions. The $E_{Abs}$ scale in panels (d) – (f) are calculated by dividing the measured MAC by the MAC for uncoated BC reported by Bond and Bergstrom (2006), which is represented by the vertical dashed lines.

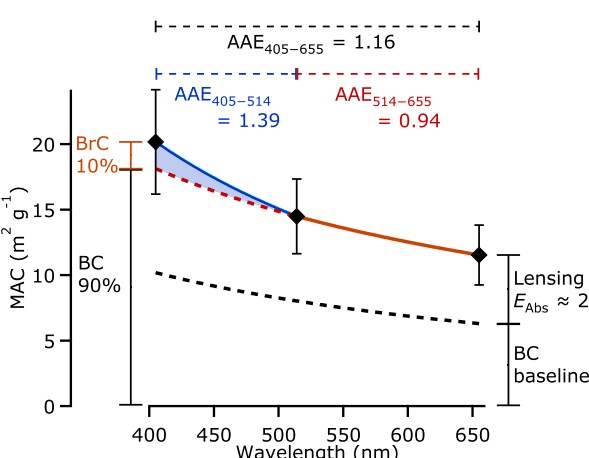

**Figure 4.** Average measured MAC as at different wavelengths, showing the calculation of the BrC absorption fraction at 405 nm. The black markers are the average CLARIFY measurements, the dashed black line is the BC absorption reported by Bond and Bergstrom (2006), the coloured solid lines represent the wavelength-dependent MAC based on the measured AAE between the two wavelength pairs, while the dashed red line represents the predicted MAC if the measured AAE from the $514 - 655$ nm wavelength pair is extrapolated to shorter visible wavelengths.





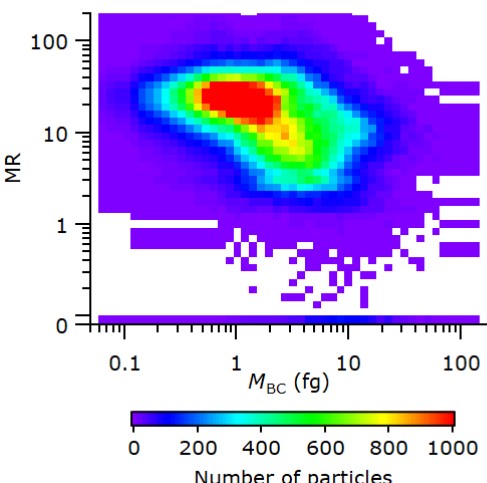

**Figure 5.** Example 2-D distribution of BC mass and mixing state, corrected for the size-dependent detection efficiency of the instrument (see Sect. S2). This distribution was taken from one straight and level run on 4 September 2017.

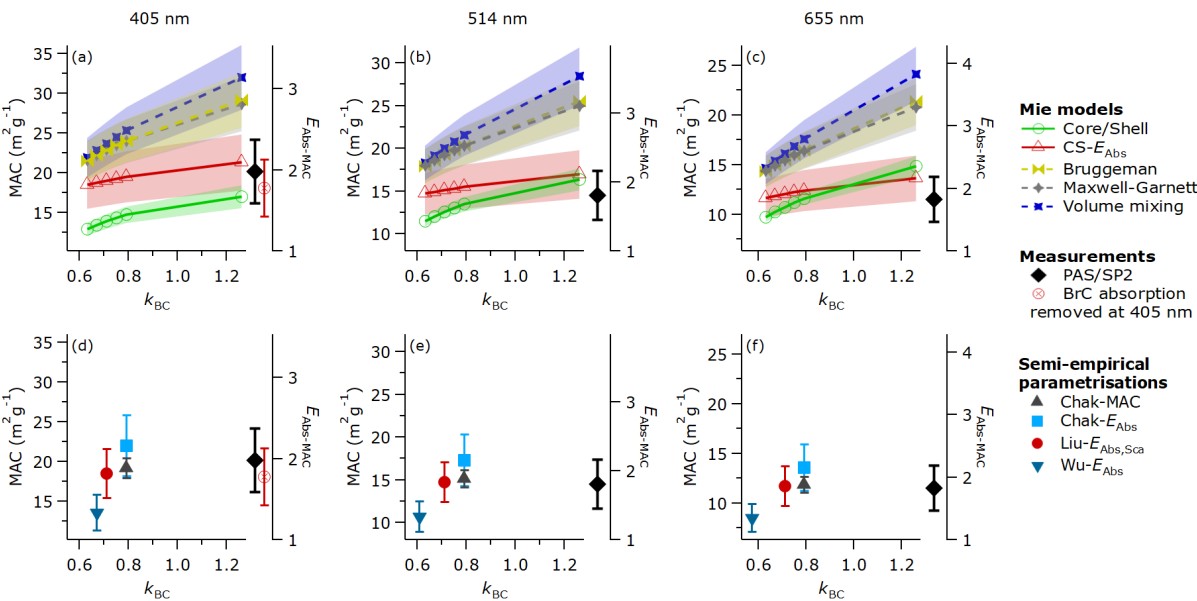

**Figure 6.** Comparison of measured MAC with values calculated using different optical models at three visible wavelengths. Panels (a) – (c) show different Mie models, whereas panels (d) – (f) show parametrisations. The right-hand axes show the equivalent values of $E_{Abs}$ by comparing values to the fresh MAC reported by Bond and Bergstrom (2006). The measured data, displayed as black diamonds to the right of each x-axis, show the weighted mean from the various straight and level runs during CLARIFY, and the errors bars are the systematic uncertainty of 19%. The red markers at 405 nm show the calculated BC MAC with no BrC, extrapolated from the longer wavelengths (see text for further details). The model and parametrisation data show the mean (unweighted) and error bars that are taken from the Monte Carlo analysis described in Sect. S3, as well as the uncertainty in the fresh MAC reported by Bond and Bergstrom (2006) for the $E_{Abs}$ parametrisations.

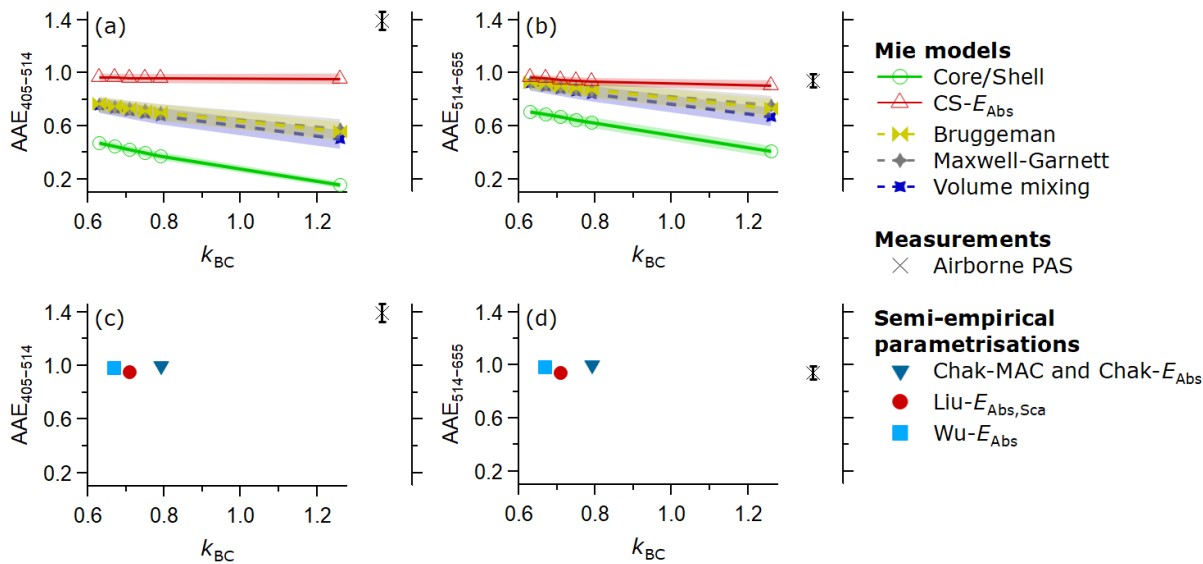

**Figure 7.** Similar to Fig. 6, but showing AAE from different optical models/parametrisations compared to CLARIFY measurements. Panels (a) and (c) show the AAE between the 405/514 nm wavelength pair, and panels (b) and (d) shows AAE between the 514/655 nm wavelength pair. The error bars are the Monte Carlo errors described in Sect. S3, and are plotted for all data points, though they are too small to see in some cases. Both schemes by Chakrabarty and Heinson (2018) have fixed AAE of exactly one, so they are listed/plotted together, and have no error bars.