# Peer review of "Absorption closure in highly aged biomass burning smoke"

_Atmospheric Chemistry and Physics, 2020_

## Referee Comment (RC1) · Anonymous Referee #1 · 13 May 2020

General Comments:

Taylor et al. present in situ airborne measurements of BC mass, microphysical BC properties, and multi-wavelength absorption in aged smoke sampled off the coast of central Africa. The dataset analyses in terms of retrieving the effective MAC and compare several models of the absorption coating enhancement. This work is an important to constraining aerosol optical properties and evaluating parameterization that may be used to more accurately model the aerosol radiative effect, specifically black carbon. This work is high quality and appropriate for ACP and should be published with minor revisions.

Specific comments:

Although technically correct, is it necessary evaluate the absorption enhancement in a quasi-single particle manner? If one assumed a log normal BC SD and an average size independent coating thickness would calculation of Eabs be significantly different?

Please consider moving section S5 and figure S5 to the main text as it is an important piece of the main conclusions of this manuscript.

Technical comments:

Page 5 line 33: Please add a reference describing the characterization of the rosemount inlet. Page 6 line 19: PCASP, for specific commercial instrumentation please state the model number and manufacturer Page 11 line 5: Please explain why only level and straight legs were used? Was the data guality better? Page 13 ine 6: The last clause of this sentence is confusing and outplace. Please remove or edit it. Page 15 line 25: This sentence is too broad. In the aged samples analyzed here, the BrC 'signal' is lower than the noise/uncertainty in the coating enhancement. However, in fresh smoke with the absorption dominated by BrC, it may be possible to extract a meaningful MAC of the OA. Page S5 line 1: Is this the same level flight leg used in the main text Fig 5? If so, please note it. Page S5, line 3: typo, 'correct' Page S11, line 20: References to the Dc distribution and Figure 5a should be update to reference figures in the manuscript. Figure 1: This figure is adequate but could be used to tell more of the story. Consider adding the approximate fire locations and arrows indicating the transport direction and time. Figure 5: Please consider added a mirrored axis on top with the spherical equ. diameter. Are other legs similar to this example? Is MR size dependent for all of the analyzed legs. Please add a sentence to the text describing the variably of this plot for the whole dataset.

---

## Referee Comment (RC2) · Anonymous Referee #2 · 25 May 2020

This study presented black carbon (BC) microphysical properties and aerosol absorption over the southeast Atlantic Ocean during the CLARFY-2017 aircraft campaign. The authors showed that BC particles have high values of mass absorption coefficient (MAC) (~20 and ~15 m2 g-1 at the wavelength of 405 and 514 nm, respectively) and absorption enhancements (~1.8) during the campaign, and these results suggest the importance of the lensing effect by coating species. The contribution of brown carbon (BrC) was estimated to be ~10% from observed absorption Angstrom exponents (AAE) at three wavelengths. The authors also made an absorption closure analysis through the comparisons of the observed MAC and AAE values with the calculated values using the Mie theory and empirical parameterizations. The authors clearly showed that the calculations by the Mie theory (homogeneous grey mixture and core-shell assumption) cannot reproduce all the observed features of MAC and AAE, while they are reproduced reasonably well by some empirical parameterizations.

The scope of this manuscript is will suited to ACP. The topic of this study is very interesting because the accurate understanding on the microphysical and optical properties of BC particles is key to improve our estimation of aerosol impacts on the global climate. The manuscript is written very well, and the uncertainties and implications of the data are discussed in detail. This manuscript should be published by ACP after revising some minor points.

Minor comments:

1) Page 1, Lines 3-4: Highly aged biomass burning plumes

The information of "4-8 days from sources" may be useful for readers.

2) Page 1, Line 12: MAC of BC

I suggest to add the values of BC MAC here (at least for the visible wavelength).

3) Page 2, Line 18:

I think 40% is too high. CMIP6 emissions (for the year 2010) are  ${\sim}10$  Tg y-1 for total BC and  ${\sim}8$  Tg y-1 for anthropogenic BC.

4) Page 4, Lines 9-13:

These sentences describe what the authors did in the manuscript. I think the authors can clarify the objectives of this study here (e.g., investigate the absorption closure between microphysical and optical properties of BC for highly-aged biomass burning plumes).

5) Page 4, Section 2.2:

Please provide the particle size ranges observed by the SP2 and PAS. The SP2 could measure most BC particles in the atmosphere? The size range of SP2 is consistent
with that of PAS? Their difference can affect the results and conclusions of this study?

6) Page 4, Lines 12-18:

Please clarify how AMS and CO data were used in this study.

7) Page 6, Line 7:

Delete "to".

8) Page 7, Line 6:

Kondo et al. (2011), which showed MMD for biomass burning plumes, can be cited here.

9) Page 7, Line 25: The ratio of observed MAC to the values by Bond and Bergstrom (2006)

I think the uncertainty in the values by Bond and Bergstrom (2006) should be considered in the Eabs estimation (1.8  $\pm$  "uncertainty" is better).

10) Page 7, Lines 22-28:

Please add MAC values at the three wavelengths to this (or related) paragraph. I think MAC values themselves are important.

11) Page 8, Lines 6-7:

The particles size ranges are consistent between SP2 and AMS? This should be considered in the uncertainty in OA MAC.

12) Section 3:

The authors should note the importance of aerosol water to MAC and Eabs in this or discussion section. The values in this study are for dry conditions, but MAC and Eabs in the real atmosphere (ambient RH) are important in evaluating their climate impacts.

13) Page 8, Lines 31-32:
It is better to describe the optical models and parameterizations used in this study briefly in the main text.

14) Page 9, Lines 14-23:

This part (steps 1-6) is not easy to understand. How about adding a figure to explain this process?

a) Please clarify steps 1-2 are theoretical calculations and steps 3-6 use observed data.

b) Please explain what is the Liu et al. correction.

c) The spherical-equivalent core in step 4 is the same as the spherical-equivalent DBC in step 3? Step 4 is to calculate shell diameter only?

d) "Convert the single-particle data to equivalent MBC and MR": I think MBC is calculated in step 3. So, step 5 is to calculate MR?

e) Step 6: this is not easy to understand unless readers read SI.

15) Page 10, Lines 9-12:

This part should be explained near the explanation of the 6 steps.

16) Page 11, Line 19:

The core/shell Mie model (green lines in Figs 6 and 7)

17) Page 11, Line 30:

Please explain briefly what is the "skin depth effect" here (though they are described in SI). Some papers should be cited.

18) Page 12, Lines 8-10:

I suggest the authors to add observed AAE for BC only to Fig 7a and 7c (like Fig 6a and 6d).

**ACPD**
19) Page 12, Lines 11-12:

Is it not possible to show the kBC dependency of MAC and AAE for the parameterizations? Please explain why the results are shown at a kBC value for each parameterization.

20) Page 15, Line 32:

RI should be changed to refractive index.

21) Text S2, Line 3:

do not "correct".

Kondo, Y., et al. (2011), Emissions of black carbon, organic, and inorganic aerosols from biomass burning in North America and Asia in 2008, J. Geophys. Res., 116, D08204, doi:10.1029/2010JD015152.

**ACPD**

---

## Referee Comment (RC3) · Anonymous Referee #3 · 31 May 2020

The authors have investigated the optical properties of black carbon (BC) and organic carbon from highly aged biomass burning plumes as part of the CLARIFY-2017 field campaign. They measure the mixing state of BC using an SP2, and the optical properties using photoacoustic spectroscopy. They use these measurements to obtain MAC, MAC enhancement, and AAE values for the aged biomass burning aerosol. These measurements are then compared to several different models for calculating biomass burning optical properties. These include coated sphere models, homogeneous grey sphere models, and more complicated aerosol optical models that account for aerosol morphology (semi-empirical models). These measurements also allow for an estimation of the contribution of brown carbon aerosol to overall absorption in aged biomass burning aerosols (10% at 405 nm). The authors conclude that all models are sensitive

to the choice of refractive index for BC. The authors also conclude that Mie models be implemented with great caution when calculating aerosol optical properties.

Major comments:

1) The authors rely heavily on SP2 measurements for most of their analysis. It would be helpful to comment on potential effects of charring of organics in the SP2 as detailed in Sedlacek et al. 2018 (Aerosol Research Letters 52:15, 1345-1350) and if these would affect any of the measurements detailed here.

2) p 14. line 12-22: The authors describe alternatives to the lensing effect of MAC and mention the possibility of externally mixed intermediate absorbers (IA) affecting total particle absorption and demonstrate that the resulting calculations do not match their observations. If possible, could the authors perform similar calculations for IA internally mixed with BC and show if such a scenario matches the values observed here. An internal mixture of IA and BC would reduce the BC MAC while also reducing the resultant AAE.

3) The main critique I have of this paper is that they provide too little detail on what makes each optical model unique. It is good that they are verifying different optical models with real world data, but one needs to be familiar with the models used for it to make sense why they give different results. I believe a little more explanation is warranted.

4) The semi-empirical models all matched the measured AAE well, and MAC values calculated using Chakrabarty and Heinson method and the Liu method matched the measured values well. MAC enhancement predicted using Liu's method matched MAC enhancement values the closest, but it is unclear why the Chakrabarty MAC enhancement did not, as they are very similar techniques. The authors reason that the enhancement calculated using the Chakrabarty and Liu methods give different results but are similar methods. The authors speculate that this has to do with morphology, did they collect any samples to image the particle morphologies?

[Figure]

5) Overall, the paper is well written but is a bit lengthy. I think perhaps the finer details in sections 4 and 4.1 could be shortened or relegated to the SI.

Other comments:

1. Section 2.2: Are there any limitations or artifacts in the instrumentation that should be mentioned or accounted for?

2. Figure 3: Should error in the MAC of BC as reported by Bond and Bergstrom account for error?

3. p. 6, line 1: Were checks put on the upper limit of the SP2 measurements as high BC concentrations can be underestimated by the instrument, or were concentrations below the upper limit of the SP2 measurement range throughout the campaign?

4. p. 8, line 8: As the OA absorption is calculated by subtracting total absorption by extrapolated BC absorption, the uncertainty propagation would also need to account for uncertainties in BC absorption measurements.

5. Page 8: add details about optical models

6. Page 9: I think the 6 step outline is going to be confusing for some, I would consider rewriting to make it more clear what is a measurement and what is a theoretical calculation

7. Page 10: There is some explanation of the Liu correction that should be moved to an earlier spot in the text

8. p 11. line 24: typo "experimental" written twice

9. p 13. line 21: It would be better to quantify the coating rather than just stating the particles to be thickly coated.

10. p 16. line 11: The line reads as if BC acts as the coating material and I think that is not the intended meaning here. Please edit the sentence to make it clear.

---

## Short Comment (SC1) · 10 Jun 2020

In their manuscript, Taylor et al. conclude that the contribution to aerosol light absorption at 405 nm by brown carbon (BrC) is roughly 10%, as inferred from the difference in the measured light absorption at that wavelength and the value extrapolated from measurements at 514 and 655 nm together with the assumption that the absorption due to black carbon (BC) over all three of these wavelengths is inversely proportional to the wavelength. We wish to point out that this is not necessarily a valid assumption. We are not suggesting that they did not measure absorption from BrC (which they discuss in more detail later in their manuscript), but merely want to state that the absorption from black carbon particles is not always inversely proportional to the wavelength; or, alternatively, that the absorption Angstrom exponent (AAE) for BC is not exactly equal

to unity.

To demonstrate that this is the case, we calculated (see Fig. 1) the AAE for the 405-514 nm and the 514-655 nm wavelength pairs for monodisperse aerosols of pure BC spheres, using the index of refraction used by Taylor et al. (2.26-1.26i). The AAE for the 405-514 nm pair increases from 1.0 for very small diameters (< ~20 nm) up to a maximum of 1.54 for 80 nm diameter particles, after which it decreases to 1.43 for 100 nm diameter particles, 0.2 for 150 nm diameter particles, and -0.19 for 200 nm diameter particles, remaining below zero for larger ones. The behavior of the 514-655 nm AAE is similar, but the diameters are shifted to larger values. For diameters less than ~90 nm, the AAE for the 405-514 nm pair is greater than that for the 514-655 nm pair, and the argument of Taylor et al. would attribute some of the BC absorption at 405 nm to BrC. Similarly, for diameters greater than ~90 nm, there would be a deficit of absorption at 405 nm.

For BC particles with associated substances (commonly referred to as coatings) the situation is perhaps more extreme. We also performed calculations for BC coated with a nonabsorbing coating in a concentric core-shell configuration, using 1.5-0i for the index of refraction of the coating, a BC core mass-equivalent diameter of 100 nm (corresponding to a mass of 0.94 fg), and a coating:core mass ratio of 20 (corresponding to a coating thickness of 104 nm, using a core density of 1.8 g/cm^3 and a coating density of 1.3 g/cm^3). Such particles are in the center of the hot spot of their 2-D distribution shown in Fig. 5 of their manuscript. For such a large coating:core mass ratio the assumption that a core-shell configuration accurately yields the absorption of the particle seems not unreasonable. The AAE for the 405-514 nm wavelength pair is 0.49, whereas that for the 514-655 pair is 1.53, neither of which is near unity. Furthermore, extrapolation of the latter AAE to 405 nm would result in less absorption than measured.

We realize that BC particles are not spheres, and perhaps not concentric core-shell configurations, and certainly not monodisperse. However, the assumption that the

AAE is identically unity for BC absorption, which is the premise of one of the arguments made by Taylor et al. to infer BrC absorption, is not necessarily true.

Ernie R. Lewis, Brookhaven National Laboratory; Arthur J. Sedlacek III, Brookhaven National Laboratory; Timothy B. Onasch, Aerodyne Research Incorporated

[Figure]

**Fig. 1.** absorbing Angstrom exponent of pure black carbon spheres

---

## Author Comment (AC1) · 21 Jul 2020

Dear Reviewers and Editor

We thank you all for your useful comments, which we have addressed below. Our responses are shown in blue text, while additions to or quotes from the manuscript are indicated by *italicised blue text*. Additionally, since the ACPD paper was submitted there has been a minor update to the photoacoustic data, which involves an update the way the background and microphone pressure sensitivity calculation was applied. Essentially rather than doing one fit per flight, an average fit for the whole dataset was used instead, with one fit for each wavelength. This has resulted in minor changes to some of the numbers for MAC and AAE but no change to the narrative or conclusions of the paper. Here is a summary of the changed numbers

| | Old | New |
|---|---|---|
| Mean MAC 405nm | 20.2 $m^2g^{-1}$ | 20.3 $m^2g^{-1}$ |
| Mean MAC 514nm | 14.5 $m^2g^{-1}$ | 14.6 $m^2g^{-1}$ |
| Mean MAC 655nm | 11.5 $m^2g^{-1}$ | 11.8 $m^2g^{-1}$ |
| Mean $AAE_{405-514}$ | 1.39 | 1.38 |
| Mean $AAE_{514-655}$ | 0.94 | 0.88 |
| Mean $AAE_{405-655}$ | 1.16 | 1.13 |
| Approx. Mean $E_{Abs}$ | 1.8 | 1.85 |
| BrC absorption fraction 405nm | 10% | 11% |
| BrC MAC 405nm | 0.27 $m^2g^{-1}$ | 0.31 $m^2g^{-1}$ |

We have also corrected an error in the caption to Figure 3 where the different panels were referred to incorrectly.

**REVIEWER #1**

Taylor et al. present in situ airborne measurements of BC mass, microphysical BC properties, and multi-wavelength absorption in aged smoke sampled off the coast of central Africa. The dataset analyses in terms of retrieving the effective MAC and com-pare several models of the absorption coating enhancement. This work is an important to constraining aerosol optical properties and evaluating parameterization that may be used to more accurately model the aerosol radiative effect, specifically black carbon. This work is high quality and appropriate for ACP and should be published with minor revisions.

**Specific comments:**

Although technically correct, is it necessary evaluate the absorption enhancement in a quasi-single particle manner?  If one assumed a log normal BC SD and an average size independent coating thickness would calculation of Eabs be significantly different?

This has been recently investigated by Fierce et al. (2020) https://doi.org/10.1073/pnas.1919723117

We have added into 4.2

*"We used a full 2-D bin scheme as absorption calculations using modal schemes, which assume a particular value of $M_{BC}$ or MR, may show significant deviations from explicit calculations (Fierce et al. 2020)"*

Please consider moving section S5 and figure S5 to the main text as it is an important piece of the main conclusions of this manuscript.

We have moved figure S5 to the main text as a new figure and incorporated the brief text into the caption.

**Technical comments:**

Page 5 line 33:  Please add a reference describing the characterization of the rose-mount inlet.

We have added a reference to the Rosemount characterisation technical note.

Page 6 line 19:  PCASP, for specific commercial instrumentation please state the model number and manufacturer

Done, but moved to section 2.2.

Page 11 line 5:  Please explain why only level and straight legs were used?  Was the data quality better?

We have added

*"The use of straight and level runs allows us to have longer averaging times (typically 5 – 15 mins), minimising statistical uncertainties, as well as negating any possibly data misalignments due to different length sample lines."*

Page 13 line 6: The last clause of this sentence is confusing and outplace. Please remove or edit it.

It now says

*"One of the key features of these African smoke plumes is their long lifetime. After several days ageing in the tropical sun, visible absorption by BrC is dwarfed by absorption by BC."*

Page15 line 25: This sentence is too broad. In the aged samples analyzed here, the BrC 'signal' is lower than the noise/uncertainty in the coating enhancement. However, in fresh smoke with the absorption dominated by BrC, it may be possible to extract a meaningful MAC of the OA.

We have clarified that this comment only refers to this dataset.

Page S5 line 1: Is this the same level flight leg used in the main text Fig 5? If so, please note it.

Noted.

Page S5, line 3: typo , 'correct'

It is now correct.

Page S11, line 20: References to the Dc distribution and Figure 5a should be update to reference figures in the manuscript.

This has been updated to Fig. S2a.

Figure 1: This figure is adequate but could be used to tell more of the story. Consider adding the approximate fire locations and arrows indicating the transport direction and time.

These have been added to Figure 1.

Figure 5: Please consider added a mirrored axis on top with the spherical equ. diameter. Are other legs similar to this example? Is MR size dependent for all of the analyzed legs. Please add a sentence to the text describing the variably of this plot for the whole dataset

We have added in the mirror axes and change the caption accordingly. We've also added

*"Equivalent distributions were generated for each straight and level run during the campaign, and the broad features were similar across all the distributions showing biomass burning aerosol"*

All the biomass burning distributions looked very similar on visual inspection. As a sanity-check we had a look at emissions from the diesel ground power unit when on the ground, and this showed just thinly coated particles across all sizes.

**REVIEWER #2**

This study presented black carbon (BC) microphysical properties and aerosol absorption over the southeast Atlantic Ocean during the CLARFY-2017 aircraft campaign. The authors showed that BC particles have high values of mass absorption coefficient (MAC) ($\sim$20 and$\sim$15 m2 g-1 at the wavelength of 405 and 514 nm, respectively) and absorption enhancements ($\sim$1.8) during the campaign, and these results suggest the importance of the lensing effect by coating species. The contribution of brown carbon (BrC) was estimated to be$\sim$10% from observed absorption Angstrom exponents (AAE) at three wavelengths. The authors also made an absorption closure analysis through the comparisons of the observed MAC and AAE values with the calculated values using the Mie theory and empirical parameterizations. The authors clearly showed that the calculations by the Mie theory (homogeneous grey mixture and core-shell assumption) cannot reproduce all the observed features of MAC and AAE, while they are reproduced reasonably well by some empirical parameterizations.

The scope of this manuscript is will suited to ACP. The topic of this study is very interesting because the accurate understanding on the microphysical and optical properties of BC particles is key to improve our estimation of aerosol impacts on the global climate. The manuscript is written very well, and the uncertainties and implications of the data are discussed in detail. This manuscript should be published by ACP after revising some minor points.

**Minor comments:**

1) Page 1, Lines 3-4: Highly aged biomass burning plumes The information of "4-8 days from sources" may be useful for readers.

*We have added this information.*

2) Page 1, Line 12: MAC of BC I suggest to add the values of BC MAC here (at least for the visible wavelength).

*We have added these values to the abstract.*

3) Page 2, Line 18:I think 40% is too high. CMIP6 emissions (for the year 2010) are$\sim$10 Tg y-1 for total BC and$\sim$8 Tg y-1 for anthropogenic BC.

*We have changed this to "open biomass burning is a major source of global BC emissions".*

4) Page 4, Lines 9-13:These sentences describe what the authors did in the manuscript. I think the authors can clarify the objectives of this study here (e.g., investigate the absorption closure between microphysical and optical properties of BC for highly-aged biomass burning plumes).

*This now says*

*"We quantify the range of values of measured MAC and AAE and investigate the absorption closure between microphysical and optical properties of BC for highly-aged biomass burning plumes."*

5) Page 4, Section 2.2:Please provide the particle size ranges observed by the SP2 and PAS. The SP2 could measure most BC particles in the atmosphere? The size range of SP2 is

consistent with that of PAS? Their difference can affect the results and conclusions of this study?

We have added in to section 2.2:

*"For BC mass measurements, the SP2 detection limits are driven by a gradual drop-off in detection efficiency for particles with BC content less than around 1 fg (102 nm mass-equivalent core diameter ($D_C$ ) (Schwarz et al., 2010), and a sharp cut-off at 143 fg (533 nm equivalent $D_C$ ), where the incandescence detector saturates. For particles that saturate the incandescence detector, we assume the BC content is 143 fg. Particles with BC content less than 1 fg are numerous but contain a negligible fraction of the total BC mass. Particles larger than 143 fg BC are rare, and by examining lognormal fits to the BC mass distribution, we estimate the uncertainty in the BC mass concentration caused by detector saturation is less than 1%. The SP2's upper cut-off diameter in terms of total particle diameter ($D_P$, i.e. the coated diameter for coated particles) is not affected by detector saturation in any practical sense, and is determined by aerodynamic limitations of particles entering the inlet, and is likely to be in the region of 1 μm. The instrument inlets are discussed further below."*

We had already stated later in Section 2.2 that the PAS samples behind a 1.3μm impactor. We now discuss a few paragraphs later the slightly different cutoff diameters of the various instruments:

*"The inboard aerosol instruments all sampled from Rosemount inlets, and the instrumental setup and detection ranges mean they all sampled a comparable size range of accumulation mode submicron aerosol (Trembath et al., 2012). The main instruments discussed here have slightly different upper cut-off diameters; 1.3 μm for the PAS, 1 μm for the AMS, and around 1 μm for the SP2. Wu et al. (2020) presents aerosol size distributions measured at ambient humidity using a passive cavity aerosol spectrometer probe (PCASP, Droplet Measurement Technologies, model SSP-200). Examination of these size distributions determined that the difference between a cut-off of 1.0 and 1.3 μm equated to 1.5% in terms of the total particle volume distribution."*

6) Page 4, Lines 12-18:Please clarify how AMS and CO data were used in this study.

We've clarified that

*"The AMS data are used to provide some context to the black carbon and optical measurements, as well as to calculate the density of the non-BC components. An in-depth discussion of the aerosol chemical composition and vertical profile is presented by Wu et al (2020)"*

and

*"The CO data are used as a measure of the amount of pollution throughout the atmospheric profile."*

7) Page 6, Line 7:Delete "to".

Done.

8) Page 7, Line 6:Kondo et al. (2011), which showed MMD for biomass burning plumes, can be cited here.

Done.

9) Page 7, Line 25: The ratio of observed MAC to the values by Bond and Bergstrom(2006)I think the uncertainty in the values by Bond and Bergstrom (2006) should be considered in the Eabs estimation (1.8±"uncertainty" is better).

We have added in uncertainties to $E_{Abs}$ throughout the paper, these are now all 1.85 +/- 0.45.

10) Page 7, Lines 22-28:Please add MAC values at the three wavelengths to this (or related) paragraph. I think MAC values themselves are important.

We have added in the average values and uncertainties to this section.

11) Page 8, Lines 6-7:The particles size ranges are consistent between SP2 and AMS? This should be considered in the uncertainty in OA MAC.

We have added in to the AMS description:

*"The AMS detection size range is determined by the transmission of the aerodynamic lens on the instrument's inlet, which has transmission efficiency near 100\% over the size range 50 – 1000 nm (Liu et al. 2007)."*

Pleas also see the previous comment about the size ranges of the PAS and SP2.

12) Section 3: The authors should note the importance of aerosol water to MAC and Eabs in this or discussion section. The values in this study are for dry conditions, but MAC and Eabs in the real atmosphere (ambient RH) are important in evaluating their climate impacts.

We have added to the end of section 5.1:

*It is important to note that all our measurements took place at dry humidities, and our modelling did not include the effects of aerosol water. Haslett et al. (2019) used a core/shell Mie model to calculate that in aged plumes measured over southern West Africa, the condensation of water at relative humidities up to 98% at the top of an aerosol layer could cause the aerosol optical depth to increase by a factor of over 1.8. Experimental studies of absorption at these high humidities are rare, though Brem et al. (2012) observed that absorption of OM generated by wood pyrolysis increased by over a factor of 2 as the RH increased from 32% to 95%, where scattering only increased by a factor of ~1.4. However, there was little change in absorption at humidities below 80%, and both absorption and scattering showed steep rises at RH greater than 90%. In our dataset these humidities were not reached throughout the bulk of the aerosol plume in the atmospheric column, as shown in Fig. 2, but they were sometimes observed in clear-sky conditions near the top of the boundary layer. The effects of high relative humidity on aerosol absorption are poorly constrained, and although we are not able to provide any further constraint from our measurements, we recommend further study on this topic.*

13) Page 8, Lines 31-32: It is better to describe the optical models and parameterizations used in this study briefly in the main text.

We have added some extra to the main text, so this now says

*"The optical models and parametrisations tested in this analysis are listed in Table 1, and described in detail in supplementary Sect. S1. We have tested the core/shell Mie model, as well as several homogeneous grey sphere models, which utilize a Mie model with a sphere of one single complex refractive index, calculated using different rules to account for the mixing between BC and non-BC components. The different mixing rules are: (i) volume mixing, where the refractive index is averaged weighted by the volume of each component; (ii) Maxwell-Garnett approximation (Markel, 2016), which considers mixing of small particles of BC dispersed throughout a non-BC host medium; and the Bruggeman mixing rule (Markel, 2016), which computes the refractive index of two components dispersed evenly within a particle. We have also tested several parametrisations of either MAC or $E_{Abs}$ (Liu et al., 2017; Chakrabarty and Heinson, 2018; Wu et al., 2018), which are based on empirical or semi-empirical fits to MAC or $E_{Abs}$ for particles with different mixing states using real and/or modelled particle data."*

14) Page 9, Lines 14-23:This part (steps 1-6) is not easy to understand. How about adding a figure to explain this process?

We have added a new figure to the supplementary section (Figure S2) showing a schematic of the SP2 mass distribution processing, while we refer the reader to this schematic in the main text.

 a) Please clarify steps 1-2 are theoretical calculations and steps 3-6 use observed data.

We have clarified in the text that *"Steps 1 and 2 are purely theoretical calculations, whereas steps 3 -- 6 involve processing of the measured data."*

 b) Please explain what is the Liu et al. correction.

We have added information concerning the Liu et al correction *"This empirical correction to the core/shell Mie model accounts for the fact that particles with MR < 3 do not scatter light at 1064nm exactly as described by the core/shell Mie model"*.

 c) The spherical-equivalent core in step 4 is the same as the spherical-equivalent DBC in step 3? Step 4 is to calculate shell diameter only?

Yes, we have rephrased this as:

*"Process single-particle data through the table to calculate the single-particle spherical-equivalent shell diameters, including the Liu et al. correction"*.

 d) "Convert the single-particle data to equivalent MBC and MR": I think MBC is calculated in step 3. So, step 5 is to calculate MR?

Yes, we have rephrased this step

*"Convert the single-particle shell/core ratio to MR and bin the data into a 2-D distribution of MR vs $M_{BC}$"*.

e) Step 6: this is not easy to understand unless readers read SI.

We have added in a reference to supplementary S2.

15) Page 10, Lines 9-12:This part should be explained near the explanation of the 6 steps.

We have added an explanation of the Liu et al correction into step 2.

16) Page 11, Line 19:The core/shell Mie model (green lines in Figs 6 and 7)

Done.

17) Page 11, Line 30:Please explain briefly what is the "skin depth effect" here (though they are described in SI). Some papers should be cited.

We have added in *"the skin depth effect prevents light interacting fully with all the light absorbing BC, as the surface of the sphere absorbs and scatters light and shields the centre, which is then less effective at absorption (Wang et al, 2015)."* We have also referenced Chakrabarty et al (2018) a few sentences later in the same paragraph.

18) Page 12, Lines 8-10:I suggest the authors to add observed AAE for BC only to Fig 7a and 7c (like Fig 6aand 6d).

Done.

19) Page 12, Lines 11-12:Is it not possible to show the kBC dependency of MAC and AAE for the parameterizations?  Please explain why the results are shown at a kBC value for each parameteri-zation.

The parameterisations developed previously by others were for specific values of $m_{BC}$, and thus they do not enable predictions with varying $m_{BC}$. We have edited the figure caption to explain why:

*"Panels (a) -- (c) show different Mie models evaluated at various values of $m_{BC}$, plotted against $k_{BC}$ on the horizontal axis. Panels (d) -- (f) show parametrisations which are plotted at the values of $k_{BC}$ at which the parametrisations were developed."*

20) Page 15, Line 32:RI should be changed to refractive index.

Done.

21) Text S2, Line 3:do not "correct".

Done.

**REVIEWER #3**

The authors have investigated the optical properties of black carbon (BC) and organic carbon from highly aged biomass burning plumes as part of the CLARIFY-2017 field campaign. They measure the mixing state of BC using an SP2, and the optical proper-ties using photoacoustic spectroscopy. They use these measurements to obtain MAC, MAC enhancement, and AAE values for the aged biomass burning aerosol. These measurements are then compared to several different models for calculating biomass burning optical properties. These include coated sphere models, homogeneous grey sphere models, and more complicated aerosol optical models that account for aerosol morphology (semi-empirical models). These measurements also allow for an estimation of the contribution of brown carbon aerosol to overall absorption in aged biomass burning aerosols (10% at 405 nm). The authors conclude that all models are sensitive to the choice of refractive index for BC. The authors also conclude that Mie models be implemented with great caution when calculating aerosol optical properties.

**Major comments:**

1) The authors rely heavily on SP2 measurements for most of their analysis. It would be helpful to comment on potential effects of charring of organics in the SP2 as detailed in Sedlacek et al. 2018 (Aerosol Research Letters 52:15, 1345-1350) and if these would affect any of the measurements detailed here.

We have added to the experimental section

*"Under certain conditions, charring can occur as weakly absorbing particles enter the SP2's laser, causing a false black carbon signal. Sedlacek et al. (2019) showed this can occur for fulvic and humic acids (BrC surrogates), but only when they had been passed through an external heated tube furnace, which we do not use in our experiment. Artificial tar balls formed through anoxic pyrolysis may also show a false rBC signal, but tar balls formed in real fires have been observed to show no detectable incandescence signal in the SP2 (Adler et al, 2019). We therefore do not consider this to be a major concern for our observations."*

2) p 14. line 12-22: The authors describe alternatives to the lensing effect of MAC and mention the possibility of externally mixed intermediate absorbers (IA) affecting total particle absorption and demonstrate that the resulting calculations do not match their observations. If possible, could the authors perform similar calculations for IA internally mixed with BC and show if such a scenario matches the values observed here. An internal mixture of IA and BC would reduce the BC MAC while also reducing the resultant AAE.

The calculations suggested by the reviewer are not easy to perform. Firstly, we would need to know the wavelength-dependence of the IA refractive index, which is currently unknown. We would also find that the answer would strongly depend on which model was used to calculate the optical properties of the internally mixed particles. For example, see our response to Dr Lewis et al. below and the new Table 2, which show large model-to-model variability even without the presence of an IA or BrC. Our absorption spectrum is only 11% different at 405nm from what we might expect from pure BC, so it is unlikely that IA can have a strong effect if they have AAE anything like those in literature.

3) The main critique I have of this paper is that they provide too little detail on what makes each optical model unique. It is good that they are verifying different optical models with real world data, but one needs to be familiar with the models used for it to make sense why they give different results. I believe a little more explanation is warranted.

*The reviewer raises a good point, which was also raised by Reviewer 2. We refer the reader to our response to comment #13 by Reviewer 2.*

4) The semi-empirical models all matched the measured AAE well, and MAC values calculated using Chakrabarty and Heinson method and the Liu method matched the measured values well. MAC enhancement predicted using Liu's method matched MAC enhancement values the closest, but it is unclear why the Chakrabarty MAC enhancement did not, as they are very similar techniques. The authors reason that the enhancement calculated using the Chakrabarty and Liu methods give different results but are similar methods. The authors speculate that this has to do with morphology ,did they collect any samples to image the particle morphologies?

*The reviewer is right to highlight our assertion that morphology may be responsible for the small differences in the predicted MAC values from the parameterisations of either Chakrabarty and Heinson (2018) or Wu et al. (2018). However, we only suggested the role of morphology as a possibility and it is beyond the scope of this paper to investigate differences in the specific simulations examined by these two previous studies. We have amended our discussion in Sect. 5.2 to read:*

*"We speculate that it may either be related to some detail of the morphology of the particles used in their simulations, or some particular details of the optical models or refractive indices used, but it is beyond the scope of this paper to make in-depth comparisons of simulations from literature."*

5) Overall, the paper is well written but is a bit lengthy. I think perhaps the finer details in sections 4 and 4.1 could be shortened or relegated to the SI.

*A lot of section 4 started off as a big supplement but several coauthors responded saying they were not familiar with the SP2 technique and didn't understand what was going on. With this in mind we have decided to leave it as is. Indeed, the SP2 analysis applied in this work is significantly different to that ordinarily applied by many others who have reported SP2 mixing state analyses (e.g. the Liu et al correction).*

**Other comments:**

1. Section 2.2: Are there any limitations or artifacts in the instrumentation that should be mentioned or accounted for?

*We have tried to incorporate these into the listed uncertainties. We have also added in the detection limits in response to reviewer #2.*

2. Figure 3: Should error in the MAC of BC as reported by Bond and Bergstrom account for error?

We have added to the figure caption:

*"The uncertainties in the average values are ±19% for MAC and ±25% for $E_{Abs}$."*

We have added into the instrumental section about the uncertainty in $E_{Abs}$:

*"We also calculate $E_{Abs}$ by dividing the measured MAC by the values reported for fresh BC by Bond and Bergstrom (2006) of $7.5 \pm 1.2 \ m^2 g^{-1}$ at a wavelength of 550 nm. The uncertainty in $E_{Abs}$ is then 25%, calculated by combining the uncertainty in our measured MAC with the range of MAC from Bond and Bergstrom (2006)."*

3. p. 6, line 1: Were checks put on the upper limit of the SP2 measurements as high BC concentrations can be underestimated by the instrument, or were concentrations below the upper limit of the SP2 measurement range throughout the campaign?

The reviewer raised a good point, we have added the following clarification to Sect. 2.2

*"The SP2 single-particle data were also examined for coincidence at high concentrations, which would cause the instrument to undercount the BC number and mass concentrations. The highest BC number concentrations measured were just below $1000 cm^{-3}$, and with this high loading 2% of particle detection windows showed coincident particles. To correct for this small bias, the coincident particles were included in the concentrations of BC mass and number concentrations, but not in the single-particle mixing state analysis, as the leading edge scattering signal can only be measured for the first particle in a measurement window."*

To be clear, the BC mass and number concentrations have not been revised since the previous version, the coincident particles were already included in the concentrations.

4. p. 8, line 8: As the OA absorption is calculated by subtracting total absorption by extrapolated BC absorption, the uncertainty propagation would also need to account for uncertainties in BC absorption measurements.

The BC concentration drops out of the calculation, like it does for the fraction of BrC absorption. To make this clearer we have defined the equations for both of these on pages 8 & 9 in Section 3, and explicitly stated that the BC concentration drops out.

5. Page 8: add details about optical models

We have now done this. Again, please see our response to comment #13 from Reviewer #2 for further details.

6. Page 9: I think the 6 step outline is going to be confusing for some, I would con-sider rewriting to make it more clear what is a measurement and what is a theoretical calculation

We have added the following clarification:

*"Steps 1 and 2 are purely theoretical calculations, whereas steps 3 – 6 involve processing of the measured data."*

Please also see our responses to reviewer 2, points 14 (a) to (e), where we have gone into more detail and included a schematic figure of the whole process.

7. Page 10: There is some explanation of the Liu correction that should be moved to an earlier spot in the text

We have included more detail in step 2

*"This empirical correction to the core/shell Mie model accounts for the fact that particles with MR < 3 do not scatter light at 1064 nm exactly as described by the core/shell Mie model"*

8. p 11. line 24: typo "experimental" written twice

Done.

9. p 13. line 21: It would be better to quantify the coating rather than just stating the particles to be thickly coated.

We have moved a later statement to the location the reviewer highlights to clearly communicate the thickness of the coating:

*"The BC particles measured during CLARIFY were universally thickly coated, with median MR values in the range 8 – 12".*

10. p 16. line 11: The line reads as if BC acts as the coating material and I think that is not the intended meaning here. Please edit the sentence to make it clear.

This now says *"coatings on BC-containing particles".*

**Ernie Lewis et al.**

In their manuscript, Taylor et al. conclude that the contribution to aerosol light absorption at 405 nm by brown carbon (BrC) is roughly 10%, as inferred from the difference in the measured light absorption at that wavelength and the value extrapolated from measurements at 514 and 655 nm together with the assumption that the absorption due to black carbon (BC) over all three of these wavelengths is inversely proportional to the wavelength. We wish to point out that this is not necessarily a valid assumption. We are not suggesting that they did not measure absorption from BrC (which they discuss in more detail later in their manuscript), but merely want to state that the absorption from black carbon particles is not always inversely proportional to the wavelength; or, alternatively, that the absorption Angstrom exponent (AAE) for BC is not exactly equal to unity.

To demonstrate that this is the case, we calculated (see Fig. 1) the AAE for the 405-514 nm and the 514-655 nm wavelength pairs for monodisperse aerosols of pure BC spheres, using the index of refraction used by Taylor et al. (2.26-1.26i). The AAE for the 405-514 nm pair increases from 1.0 for very small diameters (<~20 nm) up to a maximum of 1.54 for 80 nm diameter particles, after which it decreases to 1.43 for 100 nm diameter particles, 0.2 for 150 nm diameter particles, and -0.19 for 200 nm diameter particles, remaining below zero for larger ones. The behavior of the 514-655nm AAE is similar, but the diameters are shifted to larger values. For diameters lessthan~90 nm, the AAE for the 405-514 nm pair is greater than that for the 514-655 nm pair, and the argument of Taylor et al. would attribute some of the BC absorption at405 nm to BrC. Similarly, for diameters greater than~90 nm, there would be a deficit of absorption at 405 nm.

For BC particles with associated substances (commonly referred to as coatings) the situation is perhaps more extreme. We also performed calculations for BC coated with a nonabsorbing coating in a concentric core-shell configuration, using 1.5-0i for the index of refraction of the coating, a BC core mass-equivalent diameter of 100 nm (corresponding to a mass of 0.94 fg), and a coating:core mass ratio of 20 (corresponding to a coating thickness of 104 nm, using a core density of 1.8 g/cmˆ3 and a coating density of 1.3 g/cmˆ3). Such particles are in the center of the hot spot of their 2-Ddistribution shown in Fig. 5 of their manuscript. For such a large coating:core mass ratio the assumption that a core-shell configuration accurately yields the absorption of the particle seems not unreasonable. The AAE for the 405-514 nm wavelength pair is0.49, whereas that for the 514-655 pair is 1.53, neither of which is near unity. Further-more, extrapolation of the latter AAE to 405 nm would result in less absorption than measured.

We realize that BC particles are not spheres, and perhaps not concentric core-shell configurations, and certainly not monodisperse. However, the assumption that the AAE is identically unity for BC absorption, which is the premise of one of the arguments made by Taylor et al. to infer BrC absorption, is not necessarily true.

Ernie R. Lewis, Brookhaven National Laboratory; Arthur J. Sedlacek III, Brookhaven National Laboratory; Timothy B. Onasch, Aerodyne Research Incorporated

[Figure]

Fig. 1.absorbing Angstrom exponent of pure black carbon spheres

We thank Dr Lewis and co-commenters for their useful thoughts on our manuscript. Their concern relates to our Fig. 4 where we extrapolate the measured AAE between 514-655nm (previously 0.94, now 0.88, not unity as they state) to shorter wavelengths and use this to deduce the BrC absorption fraction at 405nm. They give some monodisperse Mie calculations that show very different AAE values to our empirical measurements in both wavelength ranges. We acknowledge that their concerns are valid, however their illustrative approach gives more extreme results than a full consideration of the BC size distribution. Firstly, when integrating the full polydisperse BC size distribution, the average AAE values settle at some average value between the extreme values they show, and this is not necessarily in the same location as the AAE of the mass or number median diameters. Secondly, when using a Mie model, particularly with a high BC refractive index, the strong skin-depth shielding effect may give low AAE values that are not observed in real particles, which are not perfectly spherical. Dr Lewis and co-commentators state that "For such a large coating:core mass ratio the assumption that a core-shell configuration accurately yields the absorption of the particle seems not unreasonable", however our results in Figure 7 show that for calculations involving AAE the core/shell Mie model does produce results that do not agree with our observations.

We re-plotted Figure 4 to estimate the BrC absorption fraction at 405 nm, but instead of using the measured $AAE_{514-655}$ to extrapolate the 655nm MAC to shorter wavelengths, we used the AAE values from the different optical models.

These are the corresponding calculated BrC absorption fractions at 405 nm:

*"Table 2. BrC absorption fraction at 405 nm calculated empirically or using the AAE from optical models. The minimum, mean, and maximum refer to the range of results from using the different values of $m_{BC}$ .*

| Model | Min | Mean | Max |
|---|---|---|---|
| *Core/shell* | *23* | *26* | *33* |
| *Bruggeman* | *13* | *15* | *21* |
| *Maxwell-Garnett* | *13* | *15* | *20* |
| *Volume mixing* | *13* | *16* | *23* |
| *Chak-MAC* | *-* | *6* | *-* |
| *Observations* | *-* | *11 +/- 2* | *-* |

"

We have added a short subsection as a new section 4.3

*"In the calculations shown in Sect. 3 and Fig. 4, we presented an estimate of the fraction of absorption at 405 nm that was due to the presence of BrC, not BC. This estimate relied on the assumption that the AAE of BC was invariant with wavelength within the visible spectrum. Mie models predict that the AAE of BC is highly dependent on the size of the particles (Lack and Cappa, 2010). When considering a polydisperse BC size distribution, much of this variability will average out, however we conducted some additional calculations to test the robustness of our empirical estimate. The MAC of BC at 405 nm was re-calculated by extrapolating the measured MAC at 655 nm, using the AAE provided by the optical models described earlier in this section, with the full 2-D mixing state of the BC-containing particles. By using only the models' wavelength dependence of absorption, this approach accounts for the over- or under-prediction of the MAC of BC at the longer wavelengths, which would otherwise have a large effect on the calculated BrC absorption fraction. As in Fig. 4, any absorption in excess to this extrapolation is ascribed to BrC. The results of these calculations are shown in Table 2. The model results were broadly consistent with our empirical calculation in that they showed that the large majority of absorption was due to BC at this short wavelength. The model-to-model variability was large, and similar in size to the calculated BrC absorption fraction."*

For reference here is the equivalent of figure 4, with the model estimates put on, but we have not included this in the revised manuscript as it looks messy and is better summarised in the table.